# Induction and regulation of reversible suspended animation in *C. elegans*

Junqiang Liu [1], Bingying Wang[1], Jonathan Leon Catrow[2], Quentinn Pearce[2], Zhijian Ji [1], Supeng Winnie Yang[1], Akash Balakrishnan[1], James E. Cox [2] & Dengke K. Ma [1,3,4] ✉

Suspended animation, a state of profound metabolic, behavioral and developmental quiescence, is a remarkable yet poorly understood stress resilience strategy in animals. Here, we describe a previously uncharacterized form of suspended animation inducible by high-population density in isosmotic liquids in *C. elegans* throughout larval development and adulthood. Transcriptomic, metabolomic, and live-cell activity reporter imaging analyses reveal striking molecular and cellular landscape changes caused by such liquid-induced suspended animation (LISA), including remodeling of gene expression programs, energy metabolites, lysosomal and mitochondrial morphology. Genetic screens identify mutants with altered stress responses and survival against LISA. While key endo-lysosomal regulators promote survival during LISA, organelle remodeling and a neuronal axis via downstream neuropeptide and cAMP/PKA signaling orchestrate behavioral awakening from LISA. Our findings define a facile paradigm for reversible SA, providing a powerful model system to uncover key molecular and cellular mechanisms governing an extreme case of reversible life arrest and dormancy.

Animals across diverse phyla have evolved remarkable strategies to endure extreme environmental stress by entering reversible states of dormancy or metabolic suppression[1]. Among these, suspended animation (SA) is one of the most profound, characterized by a nearly complete cessation of development and microscopically visible behavioral movement while preserving organismal viability[2–11]. Such states are not only biologically intriguing but also of growing interest for their potential applications in medicine, biotechnology, and space biology, where controlled metabolic arrest and reversible hibernation-like states could extend viability under limiting conditions.

Studies in the multicellular animal model organism *Caenorhabditis elegans* have provided a mechanistic foundation for understanding SA as an active, regulated response to extreme environmental stress. In seminal work by Padilla, Roth, and colleagues, complete oxygen deprivation (anoxia) was shown to induce a reversible SA state in *C. elegans*, characterized by a coordinated arrest of development, cell division, motility, and other energy-intensive processes until normoxic conditions are restored[7,12]. In embryos, anoxia triggers hypometabolism marked by reduced ATP levels and dephosphorylation of cell cycle-regulated proteins, with blastomeres arresting at specific cell cycle phases in a manner that preserves viability and structural integrity. Genetic dissection revealed that components of the spindle assembly checkpoint (e.g., *san-1*, *mdf-2*) and nucleoporin-dependent regulation of CDK-1 activity are essential for proper anoxia-induced arrest, underscoring that entry into and exit from SA require specific molecular regulators rather than passive metabolic failure[4,13,14]. Furthermore, *C. elegans* demonstrates differential responses to graded oxygen stress: while anoxia elicits behavioral responses and SA independently of the hypoxia-induced transcription factor HIF-1, milder hypoxic conditions engage HIF-1–dependent pathways[15–17]. Collectively, this literature establishes *C. elegans* as a tractable model for dissecting the cellular and genetic

[1]Cardiovascular Research Institute, University of California San Francisco, San Francisco, CA, USA. [2]Metabolomics Core Research Facility, Department of Biochemistry, University of Utah, Salt Lake City, UT, USA. [3]Department of Physiology, University of California San Francisco, San Francisco, CA, USA. [4]Innovative Genomics Institute, Berkeley, CA, USA. ✉e-mail: Dengke.Ma@ucsf.edu

regulation of SA and situates this state within a broader spectrum of environmentally induced dormancy phenomena.

In *C. elegans*, the dauer diapause has been extensively studied as another well-characterized form of developmental arrest or dormancy accompanied by distinctive morphological and behavioral adaptations[18–20]. However, dauer occurs only during a specific, programmatically defined alternative larval stage and reflects a specialized developmental pathway rather than an arrest inducible across life stages. By contrast, SA describes a more profound and inducible state of reversible arrest in development and motility. Although SA has been documented in various systems, including in nematodes and even select vertebrates[2–11], fundamental questions remain regarding molecular, cellular, and physiological regulatory mechanisms. In particular, how animals orchestrate cell quiescence, metabolic reprogramming, and neural activity to achieve a globally suspended yet fully reversible state is not well understood. The diversity of SA induction and mechanisms remains underexplored. Addressing these questions will provide critical insight into core principles of stress resilience and may enable strategies for inducing protective stasis across diverse biological contexts.

Here, we identify static liquid incubation as a previously unrecognized and robust trigger that induces SA in *C. elegans*. Under this condition, animals rapidly enter a reversible state characterized by coordinated arrest of developmental progression and motility, satisfying established operational criteria for SA. We term this induction paradigm liquid-induced suspended animation (LISA). We systematically define the physiological and temporal features of LISA and uncover key molecular mechanisms governing both survival during arrest and efficient reactivation upon return to favorable conditions. Notably, the entry pathway for LISA appears mechanistically distinct from previously characterized forms of SA or dauer formation, suggesting that multiple upstream cues can converge on a shared reversible arrest program. By leveraging this experimentally tractable system, we provide insights into how animals dynamically suppress and subsequently restore core life processes, revealing principles underlying reversible metabolic and developmental quiescence.

## Results

### Physiological specificity and reversibility of SA

In exploring various stress conditions capable of robustly inducing SA in *C. elegans*, we found that a remarkably simple "standing" procedure, i.e. allowing animals to settle at high-population density in isosmotic M9 buffer, independent of food supplementation, was sufficient to trigger this profound quiescent state (Fig. 1a). Within hours of static incubation, larvae exhibited a striking cessation of developmental progression and visible motility, hallmarks of SA, while counterpart control animals maintained in low-density (LD, *n* = 60 worms) or dispersed (in a roller) conditions continued to develop normally (Fig. 1b). We therefore identify such static liquid incubation as a robust inducer of SA and refer to this induction paradigm as liquid-induced SA (LISA) for clarity. This phenomenon was sharply dependent on population density, with near-complete penetrance observed above a defined threshold (Fig. 1c). Arrested animals retained gross morphology but did not apparently proceed through expected larval developmental transitions, including intestinal endomitosis, seam cell terminal differentiation, or dendrite branching of PVD neurons, as visualized using cell or tissue-specific reporters (Fig. 1d–e and Supplementary Fig. 1a). Notably, LISA was inducible across a wide developmental range, from early larvae to adults, in wild type and mutants for known stress-responding transcription factors, including the hypoxia-inducible factor HIF-1 and heat-inducible HSF-1, suggesting a robust and broad permissive window for entry into LISA (Fig. 1f and Supplementary Fig. 1b). Temporal analyses revealed that LISA onset was progressive and standing duration–dependent, whereas survival remained high even after extended time in LISA (Fig. 1g, h).

Importantly, we found this arrest response required isosmotic conditions: animals exposed to hypotonic media such as water displayed reduced viability and inconsistent arrest phenotypes (Fig. 1i). Medium supplementation with various amounts of bacterial food, amino acids, or vitamins did not abrogate LISA, while canonical stressors such as starvation, heat shock, or hypoxia alone were insufficient to induce it (Fig. 1j). Dauer-defective mutants, including *daf-16* and *daf-22* that disrupt dauer stress responses and pheromone pathways[21–24], respectively, entered LISA normally, indicating that this phenomenon operates independently of dauer regulatory programs (Supplementary Fig. 1b). Additionally, animals subjected to LISA during larval development exhibited normal post-recovery lifespans and can re-enter LISA upon repeated exposure (Supplementary Fig. 1c, d). After LISA and reintroduction into normal nutrient-rich media, animals exited LISA in a coordinated and timely fashion, resuming development and locomotion with temporal synchrony (Fig. 1k–l). Unlike anoxia-induced embryonic SA, LISA at multiple larval and adult stages in our study is strictly population-density dependent and SAN-1 independent (Fig. 1c, Supplementary Fig. 1g); our method is also easier to set up without using an environmental anoxia chamber. In addition, LISA is robust across common cultivation temperatures, genetic backgrounds, buffer compositions, and food availability (Supplementary Fig. 1e–h). Together, these findings define a facile, tunable and robust system for inducing and reversing LISA in *C. elegans*, establishing a tractable platform for dissecting mechanisms of reversible metabolic suppression.

### Molecular and cellular landscape profiling of SA

To elucidate the molecular landscape associated with LISA in *C. elegans*, we performed transcriptomic profiling of animals subjected to high-density isosmotic LISA. RNA-seq analysis revealed extensive transcriptional reprogramming, with a substantial number of genes either up- or downregulated in response to LISA (Fig. 2a and Supplementary Data 1). Intriguingly, among the most robustly LISA-induced transcripts were members of the small heat-shock protein family, particularly the *hsp-16* cluster, whose upregulation was sharply localized to their genomic cluster locus without affecting adjacent genes (Fig. 2b). This transcriptional induction was corroborated by *hsp-16p::GFP* transcriptional and *hsp-16p::hsp-16::mStayGold* translational reporter imaging, which showed minimal GFP or mStayGold abundance during LISA but a dramatic increase following recovery, suggestive of a delayed protein translation, rather than contemporaneous stress response (Fig. 2c and Supplementary Fig. 2a–j). Consistently, this induction was abrogated by *hsf-1* RNAi, implicating the canonical heat-shock transcription factor HSF-1 in this response[25–27] (Supplementary Fig. 2d). The degree of *hsp-16* activation was modulated by population density, echoing the physiological parameters that define LISA itself (Supplementary Fig. 2e). RNAi knockdown of the genes encoding the *C. elegans* homologs of known thermogenic regulators, including UCP-4 and SCA-1 did not affect *hsp-16* activation, whereas lower environmental temperature progressively decreased it (Supplementary Fig. 2f, g). In addition, the supernatant from LISA animals at high density was insufficient to induce LISA or *hsp-16* reporters of animals at low density (Supplementary Fig. 2k–m), arguing against a stable diffusible pheromone that transduced the population density signal. Together, these findings delineate a distinct LISA-induced transcriptional state, marked by stress-responsive gene expression changes, likely reflecting the organism's dormant and physiologically primed state.

To further characterize the physiological and metabolic states underpinning LISA, we employed a panel of fluorescent reporters and metabolomic analyses. Systematic imaging of organelle-specific live reporters revealed that LISA triggers pronounced remodeling of the mitochondrial network, favoring vesicular over fused forms, as visualized by mitochondrial matrix-targeted MAI-2::GFP (Fig. 2d, e). This

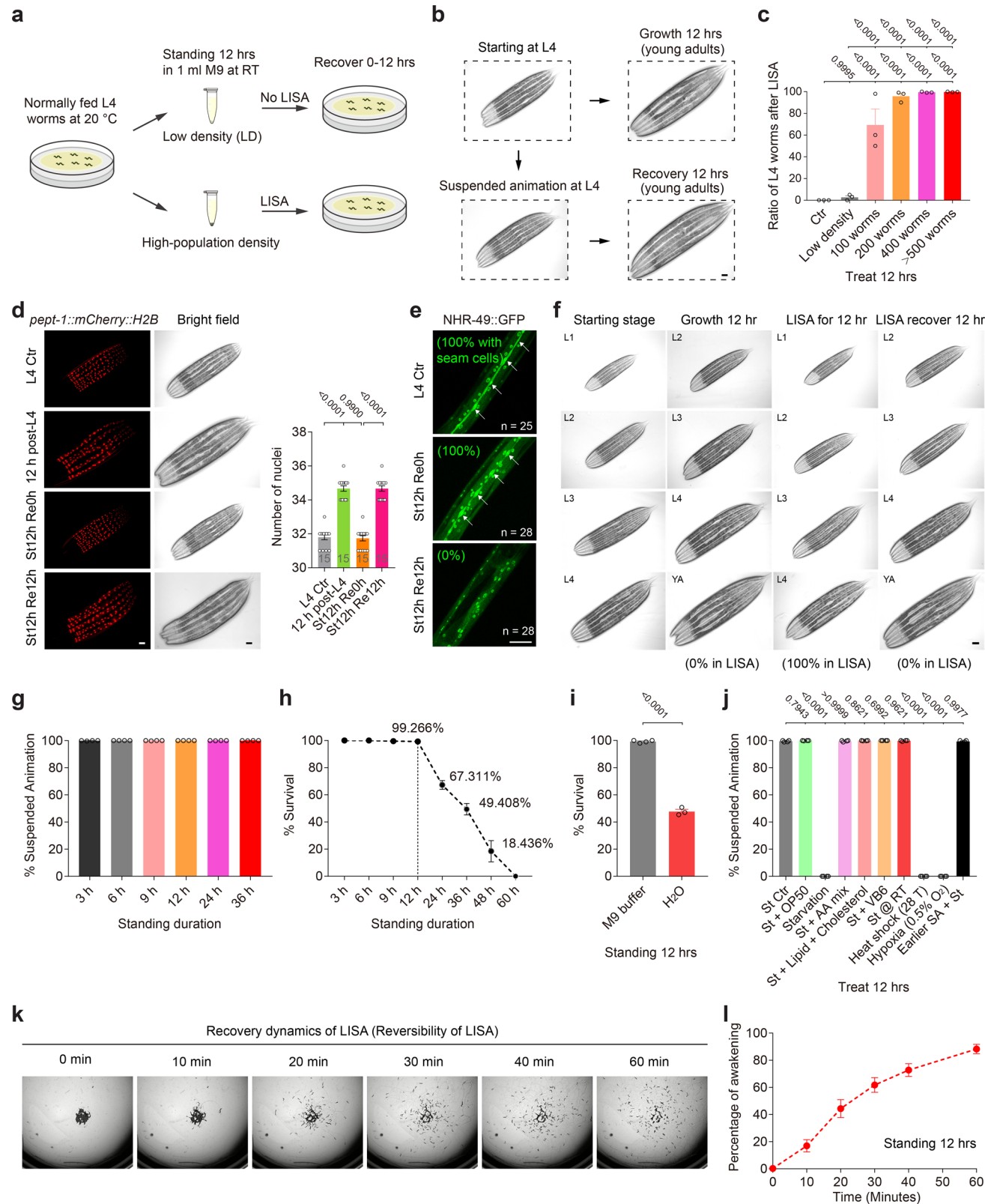

architectural shift was accompanied by a notable reduction in steady-state mitochondrial calcium levels, indicated by GCaMP-based sensors in body wall muscle cells, consistent with a hypometabolic mitochondrial state. We used liquid chromatography-mass spectrometry (LC-MS) based metabolomic profiling to further identify major energy metabolites differentially regulated by LISA (Supplementary Data 2; see Methods). Principal component analysis (PCA) of metabolites,

including ATP, ADP, AMP, acetyl-CoA, acetyl-carnitine, and various redox intermediates, showed clear segregation between LISA and control animals, indicating a comprehensive rewiring of cellular bioenergetics (Fig. 2f). We identified differential abundance of multiple metabolites critical to energy and redox homeostasis, including LISA-downregulated acetyl-CoA and ratio of NADH/NAD + , reinforcing the notion of metabolic suppression and adaptation (Fig. 2g and

**Fig. 1 | High-population density in isosmotic liquids induces suspended animation in *C. elegans*. a** Schematic of the experimental setup for inducing liquid-induced suspended animation (LISA) in *C. elegans* larvae and adults under high-population density in isosmotic liquid conditions. The diagram was generated using Adobe Illustrator (2020). **b** Representative brightfield images showing cessation of developmental progression during LISA (L4 to L4) but not control (L4 to YA) (*n* = 20 worms in each of three independent experiments). **c** Quantification of the percentage of LISA ($n$ = 50 animals per condition were observed, $N$ > 3 trials) at different population densities. Ctr: animals maintained on standard solid NGM plates without liquid immersion. *n* = 60 worms for the LD group. **d** Representative fluorescence and brightfield images showing LISA-induced arrest in the proliferation of intestinal cell nuclei, numbers of which were quantified by nuclear histone H2B-marked GFP (*n* = 15 worms in each of three independent experiments). **e** Representative fluorescence images showing LISA-induced arrest in seam cell differentiation into hypodermal cells, marked by disappearance of seam cell-expressed NHR-49::GFP (Top to bottom, *n* = 25, 28, and 25). **f** Representative brightfield images showing LISA-induced cessation of developmental progression at various stages indicated (*n* = 15 worms per condition, *N* > 3 trials). **g** Quantification of the percentage of LISA (*n* = 50 worms per condition, *N* > 3 trials) with different standing durations indicated. **h** Quantification of the percentage of survival after different standing durations indicated (*n* = 400 worms per condition in each test). **i** Quantification of percentage of survival after LISA in isosmotic M9 buffer or water only (Left to right, *n* = 400 worms per condition, with four biological replicates for M9 buffer group and three for $H_2O$ group). **j** Quantification of the percentage of LISA (*n* = 50 worms per condition, *N* > 3 trials) with different medium supplementation conditions indicated. St Standing, AA Amino acid, VB6 Vitamin B6, RT room temperature, T Temperature. **k** Representative brightfield images showing recovery following LISA. **l** Percentage of animals awakening at different times upon transfer to fresh media (*n* = 150 worms per condition, *N* > 3 trials). Data show mean ± s.e.m. **c, d, g–j, l** One-way ANOVA with Tukey's multiple comparison test (**c, d**) or Student's two-tailed unpaired *t* test (**i, j**) were used. *P* values are indicated. Scale bars, 50 μm (**b, d–f**).

Supplementary Data 2). Additionally, steady-state ATP and its dephosphorylated form AMP/adenosine were slightly up- and down-regulated, respectively, reflecting overall reduced bioenergetic demands under LISA (Fig. 2g). Complementary gas chromatography-mass spectrometry (GC-MS) metabolomic profiling of major glycolytic and TCA cycle metabolites and amino acids further supported a stress-adaptive state and hypometabolic signature characterized by elevated succinate and lactate levels (Supplementary Fig. 3 and Supplementary Data 3). Collectively, these data reveal a coordinated cellular and physiological program during LISA that conserves energy, dampens mitochondrial activity and metabolism, and primes the organism for reactivation upon stress release.

### Stress-responsive transcriptional and auto-lysosomal programs facilitate survival from SA

To uncover genetic regulators that modulate LISA and organismal survival during this profound quiescent state, we conducted forward genetic screens using ethyl methanesulfonate (EMS) mutagenesis to isolate mutants that either fail to undergo LISA by progression from L4 to adults or undergo LISA but with aberrant activation of the *hsp-16p::GFP* reporter and increased viability under prolonged LISA. Though we did not obtain fertile mutants that completely failed to undergo LISA so far, we isolated multiple mutants with constitutive *hsp-16*p::*GFP* activation and enhanced viability after prolonged LISA (Fig. 3a). Genetic mapping, complementation test and whole-genome sequencing identified causal mutant alleles in *daf-21*, encoding the Hsp90 ortholog[28], and *lin-61*, a chromatin regulator[29] (Fig. 3a and Methods). We confirmed that backcrossed *daf-21* mutants exhibited *hsp-16* expression under baseline conditions and showed further exaggerated induction after LISA, coinciding with enhanced survival following extended periods of arrest (Fig. 3b, c). These phenotypes required the heat-shock transcription factor HSF-1, as RNAi-mediated knockdown suppressed *hsp-16p::GFP* upregulation in both *daf-21* and *lin-61* backgrounds (Fig. 3d). HSF-1 and the FOXO transcription factor DAF-16 can cooperatively mediate stress-induced gene expression to promote resilience[25,30,31]. Although *hsf-1(sy441)* single reduction-of-function mutants[32] showed largely normal LISA survival, double gene manipulations revealed synergistic roles for HSF-1 and DAF-16 in LISA survival (Fig. 3e), suggesting they act in coordination and convergent pathways to promote LISA resilience.

Beyond transcriptional regulation, cellular organelle remodeling processes appeared integral to LISA survival. HSF-1 and DAF-16 can form a co-regulator complex to promote stress resilience and longevity through auto-lysosomal pathways[33–35]. We next used live imaging of an intestinal lysosomal reporter (*nuc-1::mCherry*) to monitor LISA-induced lysosomal dynamics and found that LISA triggered marked tubulation of the lysosomal network (Fig. 3f), a morphological signature often associated with stress responses and increased degradative capacity[33–35]. Genetic disruption of key auto-lysosomal pathway components[36–38], including *vps-15* (autophagosome formation), *vps-45* (autolysosome fusion), and *hpo-27* (lysosomal fission), strongly compromised survival, highlighting the functional importance of auto-lysosomal dynamics in sustaining viability during LISA (Fig. 3g). In contrast, despite robust LISA-induced mitochondrial remodeling, including altered morphology across multiple tissues (Supplementary Fig. 4a–f), mutants defective in key mitochondrial fusion and fission genes (*fzo-1*, *mtp-18*, *drp-1*)[39–41] displayed little or slightly increased LISA survival (Supplementary Fig. 4g), suggesting that mitochondrial morphologic remodeling may be a correlative rather than crucial feature of LISA resilience. Collectively, these findings reveal a critical axis of stress-responsive transcriptional programs and autophagic-lysosomal remodeling that jointly enable *C. elegans* to withstand prolonged metabolic stasis during LISA (Fig. 3h).

### Neural circuitry orchestrates behavioral awakening from SA

To delineate the neural substrates governing the transition from LISA to behavioral reactivation, we conducted a comprehensive screen across strains perturbed in key neurotransmitter systems, neuropeptides, and defined neuronal subtypes implicated in sensory perception and sleep regulation[42–46] (Fig. 4a and Supplementary Fig. 5a). This analysis revealed that the AFD sensory neurons and their postsynaptic partners AIY interneurons are critical for promoting timely awakening (defined as conspicuous locomotion from quiescent behavioral state in LISA upon transfer to normal plate culture conditions, see Methods), whereas the sleep-active RIS interneuron likely exerts an antagonistic role by suppressing arousal. In parallel, we found that neuropeptide signaling via PDF and its receptor PDFR-1 was indispensable for coordinating behavioral recovery from LISA, supporting their previously established roles in promoting behavioral "roaming"[47–51]. Notably, single-cell RNA-seq datasets[52–54] revealed enriched expression of *gcy-8*, a guanylyl cyclase, in AFD neurons, and *pdf-1* in AIY and a few other neurons, supporting their respective roles in a molecularly defined circuit for awakening regulation (Fig. 4b). Indeed, time-lapse imaging of behavioral awakening showed a time-dependent delay of recovery in AFD-ablated or PDF-deficient animals, while ablation of RIS resulted in premature awakening (Fig. 4c and Supplementary Fig. 5b). These behavioral phenotypes suggest a neural antagonism between arousal-promoting and sleep-maintaining pathways in modulating the exit from LISA.

To monitor neural activity dynamics during the transition out of LISA, we employed pan-neuronal and cell-specific calcium imaging. While global neuronal activity remained largely unchanged during LISA and recovery, we found that both AFD and AIY neurons exhibited marked silencing during LISA, followed by a progressive rise in calcium levels shortly after reintroduction to normal feeding conditions. This coordinated reactivation indicates early engagement of these neurons

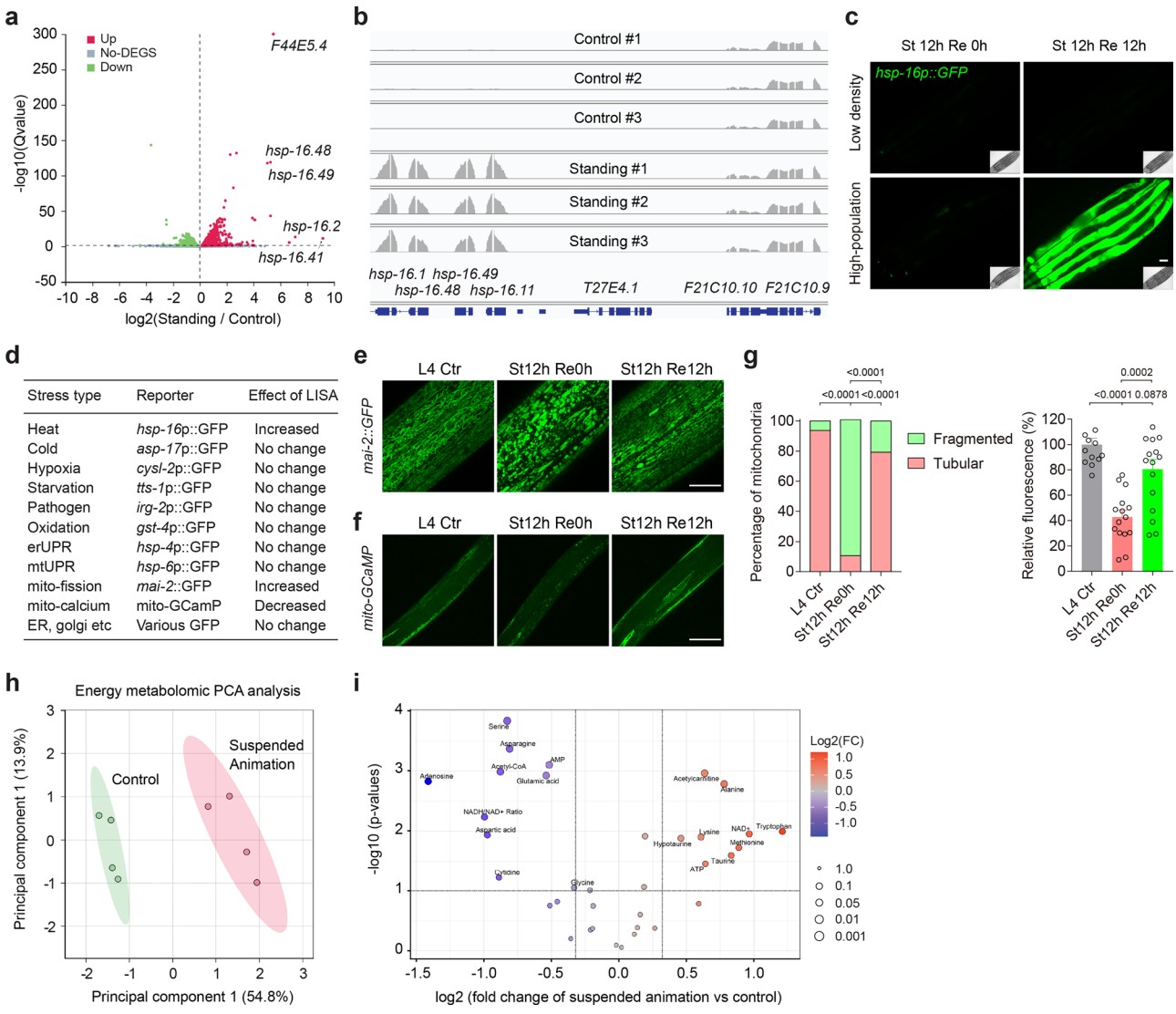

**Fig. 2 | Transcriptomic, metabolomic and imaging analyses of LISA. a** Volcano plot for RNA-seq analysis of LISA and control animals (three biological replicates were included for each treatment), showing differentially expressed genes. Differential gene expression analysis of RNA-seq was performed using the DESeq2 package. **b** RNA-seq read mapping at the *hsp-16* locus, showing drastic upregulation of the *hsp-16* family genes but not their neighbors, after recovery from LISA. **c** Representative fluorescence images of *C. elegans* hsp-16p::GFP transcriptional reporters (driven by the common *hsp-16* promoter), confirming drastic upregulation by LISA after recovery from LISA, but not during LISA (*n* = 20 worms in each of three independent experiments). **d** Table listing fluorescent reporters, with LISA effects indicated, for various stress-responding pathways, cellular organelle activities and morphologies (*n* = 15 worms in each of three independent experiments). **e**–**g** Representative confocal images of *C. elegans* expressing mitochondrial GFP

(*n* = 5 worms in each of four independent experiments) or muscle GCaMP reporters (Left to right, *n* = 17, 18, and 20 worms), showing LISA-induced mitochondrial network remodeling towards increased fission and reduction in mitochondrial calcium levels, reflecting a mitochondrial hypometabolic state. Data show mean ± s.e.m. **h** Principal Component Analysis (PCA) of energy and mitochondrial metabolites profiled, including ATP, ADP, AMP, and redox metabolites, showing well-separated clusters in control and LISA animals (Four biological replicates were included for each treatment). **i** Volcano graph to visualize metabolites analyzed by LC-MS with *P* value on the *y* axis, with fold change (FC) on the *x* axis. Differential metabolite abundance was evaluated using two-sided unpaired Student's t-tests with Benjamini−Hochberg false discovery rate correction (**i**). One-way ANOVA with Tukey's multiple comparison test was used in (**g**). All tests were two-sided. *P* values are indicated. Scale bars, 50 μm (**c**, **e**, **f**).

during awakening (Fig. 4d and Supplementary Fig. 5c, d). cAMP and PKA signaling mediates PDF/PDFR-1 signaling and constitute evolutionarily ancient regulators of awakening in animals[43,55–57]. To test whether intracellular cAMP/PKA signaling mechanisms were sufficient to modulate awakening, we manipulated cAMP levels via optogenetic activation[55] and genetic gain-of-function of adenylyl cyclase (*acy-1*)[58]. Both approaches led to precocious behavioral reactivation from LISA (Fig. 5). However, LISA-induced silencing and reactivation of AFD GCaMP activities were unaltered in PDF-1/PDFR-1 signaling-deficient or cAMP-activated *pde-4* mutant animals (Supplementary Fig. 5e), placing peptide/cAMP action downstream or in parallel to AFD. In addition, we found that loss of PDE-4 or the RIS neuron accelerated awakening even

in the absence of AFD or AIY neurons (Supplementary Fig. 6), arguing against the AFD-AIY neuronal axis acting downstream of cAMP or the RIS neuron. These results converge on a model in which AFD/AIY neurons act with downstream PDF/PDFR-1 neuropeptide signaling and subsequent cAMP/PKA activation to drive awakening via the motor circuit, while RIS neurons likely provide inhibition to maintain behavioral quiescence. Together, these findings uncover a modular neural circuit and intracellular signaling logic that governs the coordinated resumption of behavioral locomotion following LISA via cAMP/PKA signaling conserved in animals.

To determine whether the mitochondrial and lysosomal dynamics observed during LISA are functionally linked to recovery

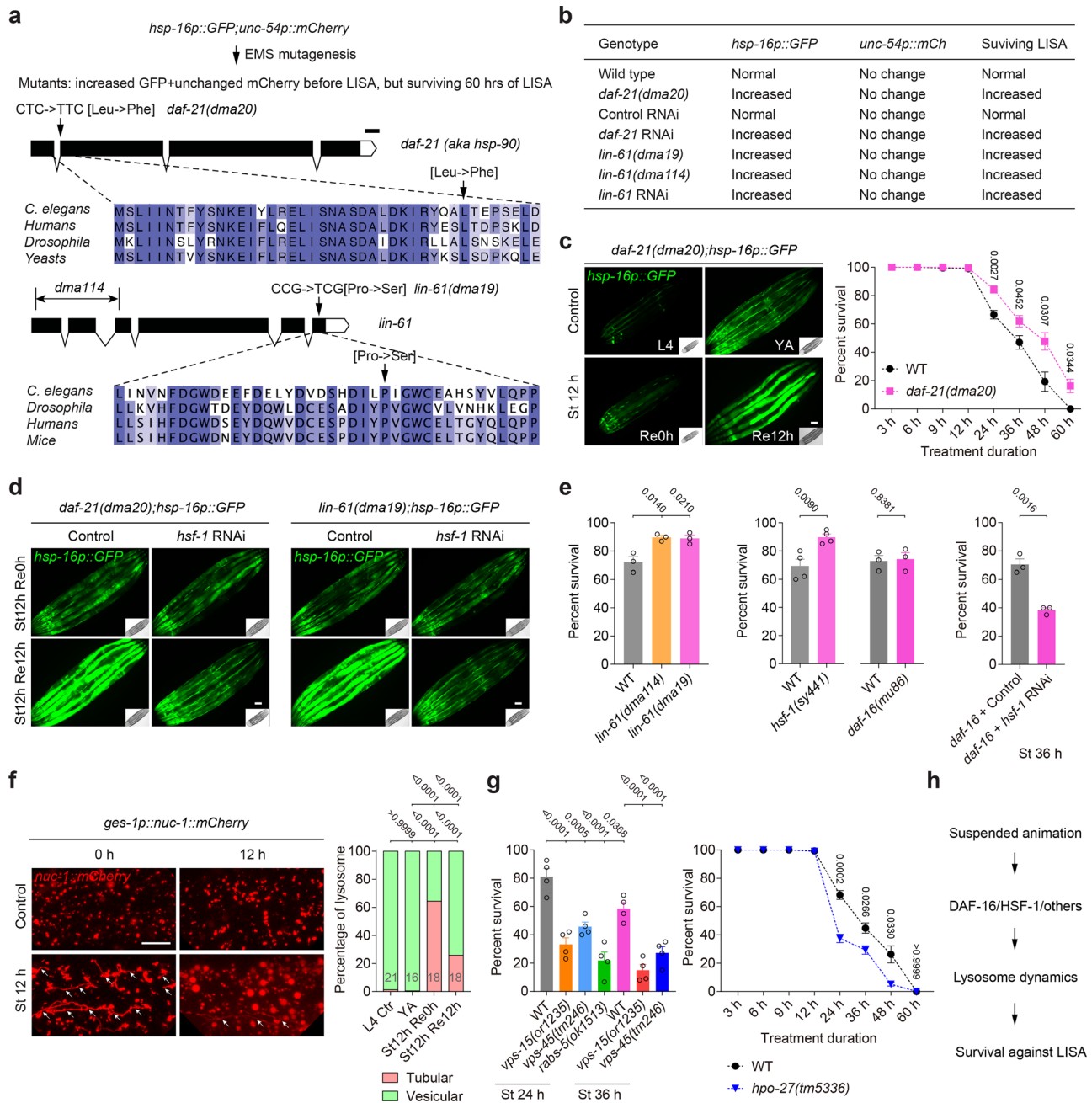

**Fig. 3 | Role of stress-responsive pathways in promoting survival during LISA.**
**a** Schematic of EMS screens for *hsp-16p::GFP* activators and LISA survivals, resulting in *daf-21* and *lin-61* mutations. **b** Representative fluorescence images of L4 stage *C. elegans daf-21* mutants with baseline constitutive *hsp-16p::GFP* expression and further enhanced response to LISA (for GFP reporter assays, *n* = 15 worms; for survival assays, *n* > 500 worms per condition, from three independent experiments). **c** Quantification of survival rates of *C. elegans* under different durations of standing, showing enhanced survival rate of *daf-21* mutants after prolonged LISA (for GFP reporter assays, *n* = 15 worms; for the survival assays, *n* = 400 worms per condition, from 3 independent experiments). **d** Representative fluorescence images of young adult-stage *C. elegans daf-21* and *lin-61* mutants with LISA-induced *hsp-16p::GFP* suppressed by *hsf-1* RNAi (*n* = 15 worms in each of 3 independent experiments). **e** Quantification of survival rates of *C. elegans* wild type and *daf-21*, *lin-61*, *hsf-1*, and *daf-16* mutants with control and RNAi against *hsf-1*, showing synergistic effects of DAF-16 and HSF-1 in surviving LISA (*n* > 500 worms per condition in each treatment). **f** Representative fluorescence images of *C. elegans* lysosomal marker *nuc-1::mCherry*, showing increased lysosomal tubulation by LISA. Quantification of lysosomal tubulation was shown (Left to right, *n* = 21, 16, 18, and 18). **g** Quantification of survival rates of *C. elegans* wild type and autophagy-lysosomal *vps-15*, *vps-45*, *rabs-5* and *hpo-27* mutants, showing key roles of lysosomal dynamics downstream of HSF-1/DAF-16 in surviving LISA (*n* = 400 worms per condition in each test). **h** Schematic model showing key processes activated by LISA to promote LISA survival. Data show mean ± s.e.m. **c**, **e**, **g** Student's two-tailed unpaired *t* test (**c**, **e**, **g**) was used to compare mutant data with WT, and one-way ANOVA followed by Tukey's multiple comparison test (**f**) to compare test versus control groups. All tests were two-sided. *P* values are indicated. Scale bars, 50 μm (**c**, **d**, **f**).

from LISA, we examined behavioral awakening in mitochondrial and lysosomal mutants. Mutations in the genes *eat-3* and *fzo-1*, essential for mitochondrial fusion caused striking defects in the LISA-induced change of mitochondrial morphology (Supplementary Fig. 4).

Correspondingly, we found that *eat-3* and *fzo-1* mutants also exhibited markedly slower awakening from LISA (Supplementary Fig. 7). Similarly, endo-lysosomal gene mutations in *vps-15*, *vps-45*, *rabs-5* and *hpo-27* led to slower awakening phenotypes (Supplementary Fig. 7).

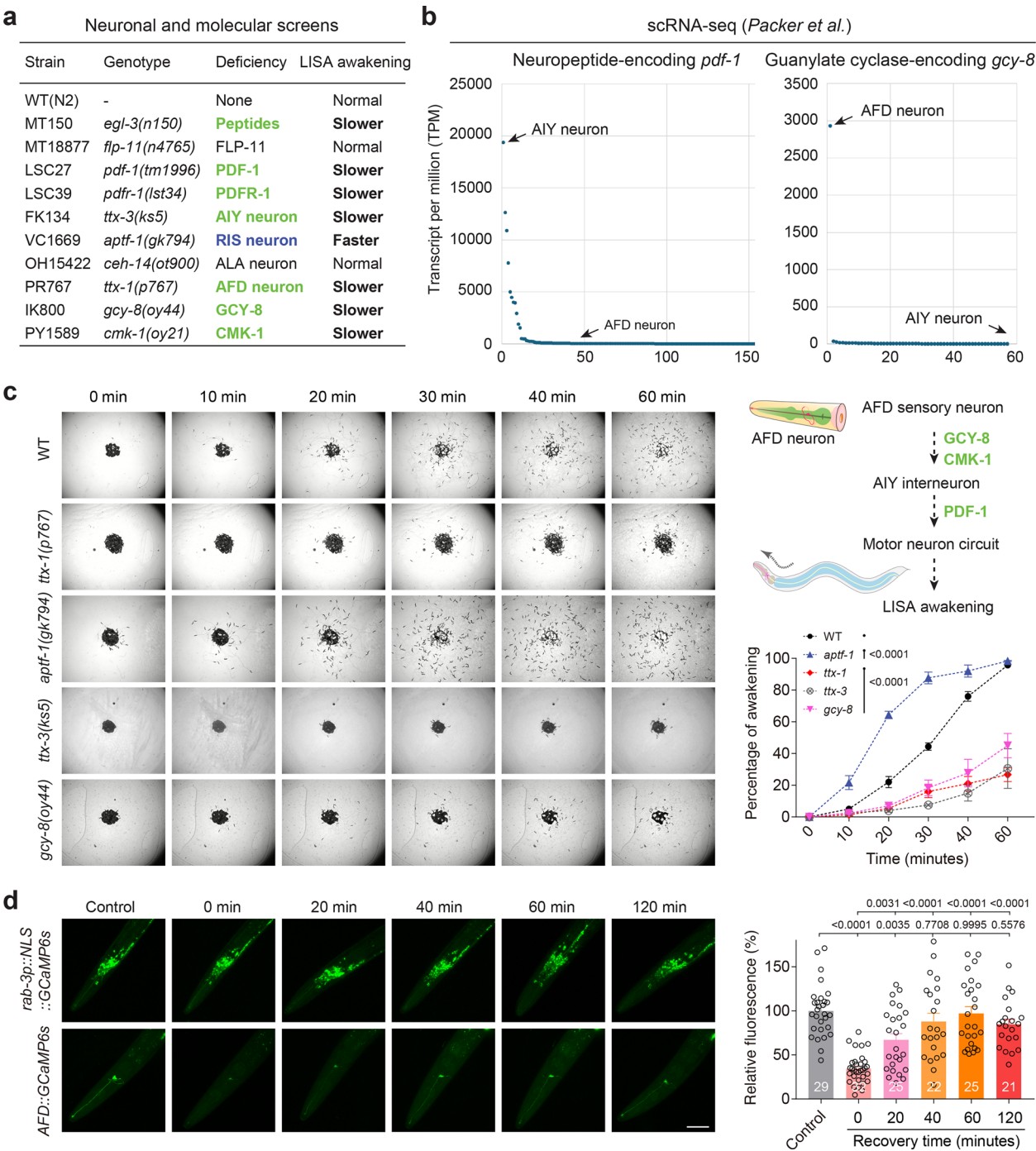

**Fig. 4 | Neural regulation of behavioral awakening upon exit from LISA. a** Table listing a panel of strains testing the importance of neurotransmitters, neuromodulators, sensory and sleep-regulating neurons in regulating the behavioral awakening from LISA (see also Supplementary Fig. 5). Such candidate screens identify essential roles of AFD and AIY neurons, neuropeptide PDF and its receptor PDFR-1 in promoting awakening from LISA, and sleep-active RIS neurons in suppressing awakening from LISA. **b** Single-cell RNA-seq dataset showing enriched expression of *pdf-1* in AIY neurons and *gcy-8* in AFD neurons. **c** Time-serial brightfield images and population quantification showing locomotion behavioral recovery post-LISA in wild type and various strains (AFD or RIS-less animals, *pdf-1*, *gcy-8* mutants), indicating differentially altered awakening behaviors after LISA (*n* = 150 worms in each of 3 independent experiments). The diagram was generated using Adobe Illustrator (2020). **d** Representative GCaMP fluorescence images and quantification for time-serial calcium levels of all neurons versus AFD neurons during LISA exit in control and LISA animals (0-60 min after LISA) (Left to right, *n* = 29, 32, 25, 22, 25, and 21). Data show mean ± s.e.m. **c**, **d** Two-way ANOVA with Bonferroni post-test was used to compare mutant data with WT (**c**). One-way ANOVA with Tukey's multiple comparison test was used in (**d**). *P* values are indicated. Scale bars, 50 μm (**d**).

By contrast, mutations in the genes key for mitochondrial fission (*mtp-18* LOF, *drp-1* LOF) or mitophagy (*dct-1* LOF, *dct-1* GOF, *pdr-1* LOF)[59,60] did not apparently affect behavioral awakening post-LISA (Supplementary Fig. 7). Collectively, these results indicate that endo-lysosomal dynamics is functionally important for not only surviving but also awakening from LISA, whereas mitochondrial fusion is selectively required for efficient behavioral reactivation from LISA, potentially by sustaining systemic energy homeostasis needed for the timely engagement of neuro-sensory and arousal-promoting circuits.

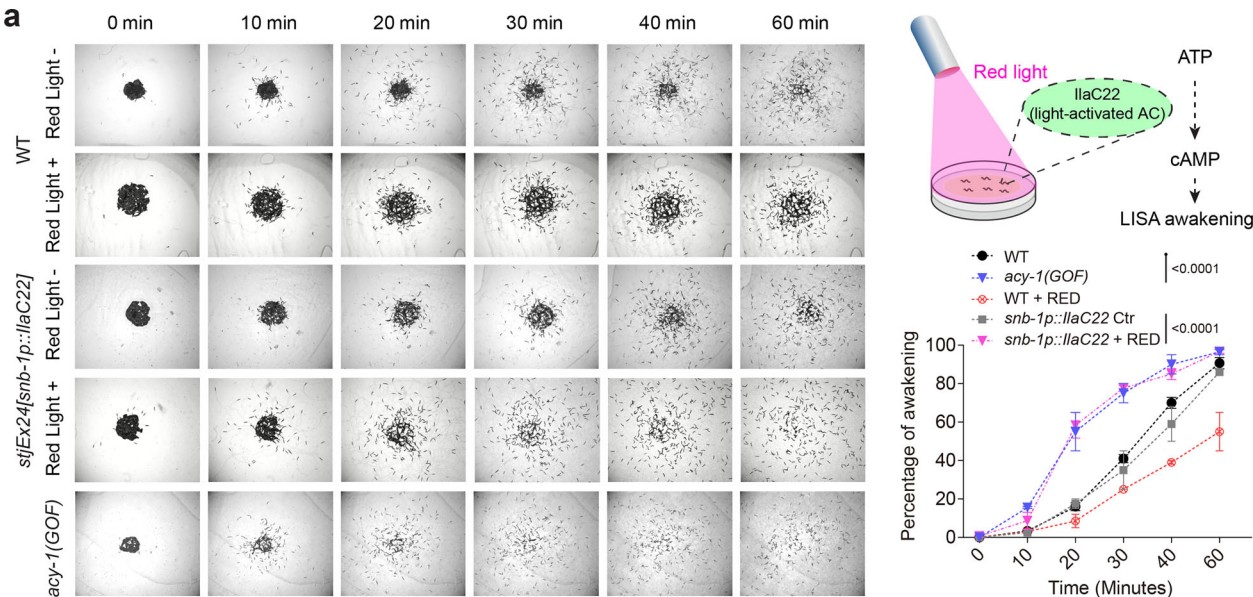

**Fig. 5 | Acute elevation of cAMP accelerates behavioral awakening from LISA.** Representative time-series brightfield images and population quantification of locomotor recovery following exit from LISA in wild-type animals, animals subjected to optogenetic activation via red-light exposure (*snb-1p::Ilac22*), and *acy-1* gain-of-function (GOF) mutants with constitutively elevated cAMP levels. Acute optogenetic activation or genetic elevation of cAMP significantly accelerates awakening from LISA ($n = 150$ worms per group in each of three independent experiments). The diagram was generated using Adobe Illustrator (2020). Data are presented as mean ± s.e.m. Statistical significance was assessed using two-way ANOVA with a Bonferroni post hoc test to compare animals with or without acute cAMP elevation. All tests were two-sided. Exact *P* values are indicated.

## Discussion

Our study establishes a robust and physiologically tractable paradigm of reversible liquid-induced SA in *C. elegans*, induced by a deceptively simple yet highly reproducible condition: settling at high-population density in isosmotic liquid. This LISA diverges from previously characterized developmental arrest states, such as anoxia-induced SA or dauer. Unlike anoxia-induced SA, our method of inducing LISA does not require a cumbersome environmental anoxia chamber, elicits a metabolically distinct bioenergetic signature, and involves high-density population crowding. The density requirement suggests a mechanism of a quorum-like, population-density–dependent regulatory process[61–63] that await further studies. Dauer occurs only during specific larval stages and involves extensive stage-specific morphogenetic remodeling. Accordingly, LISA in our study does not require specific genes, including *san-1* and *daf-22* involved in anoxia-induced SA and dauer, respectively. LISA represents a reversible pause of organismal life that largely preserves structural integrity while orchestrating profound molecular, cellular, and behavioral functional reconfigurations. The ability of animals across larval and adult stages to enter and recover from LISA with minimal detrimental sequelae underscores a remarkable plasticity in the face of environmental constraint. While the mechanism of LISA induction remains not fully elucidated, our multi-omic profiling and perturbation results together reveal that LISA is not merely a passive shutdown but a highly regulated physiological state marked by stress-responsive gene expression program, energetic reconfiguration, and neuronal orchestration, suggesting the existence of an evolutionarily conserved logic for engaging protective stasis.

Transcriptomic profiling and live reporter analyses of animals undergoing LISA reveal a molecular program dominated by the upregulation of small heat-shock proteins, particularly *hsp-16* family members. This response is driven by HSF-1–dependent transcription during arrest and by post-arrest protein translation, together supporting a protective mechanism that buffers the transition from dormancy to active metabolism and mitigates proteostatic stress during reactivation. Reduced mitochondrial calcium levels and mitochondrial

morphological changes further reflect a coordinated entry into hypometabolic stasis, a feature observed across evolutionarily distant organisms that endure extreme environmental duress. Notably, while mitochondrial remodeling is prominent, our data suggest it is dispensable for LISA survival per se, whereas transcriptional and autophagic-lysosomal pathways, particularly those downstream of HSF-1 and DAF-16, are indispensable. These findings echo prior observations that lysosomal dynamics and proteostasis networks are critical to longevity and stress resilience[64–66], reinforcing their broader relevance in metabolic arrest biology.

Our genetic screens illuminate additional layers of regulation that confer stress resilience during LISA. Mutant alleles in *daf-21* (Hsp90) and *lin-61* (a chromatin-associated factor) confer heightened *hsp-16* expression and improved viability, implicating molecular chaperone networks and repressive chromatin in modulating the threshold for stress response activation. DAF-21/Hsp90 is a central regulator of protein homeostasis[25,30], buffering misfolded and aggregation-prone proteins and stabilizing key signaling components required for stress response pathways. Its well-established role in modulating HSF-1 activity and broader chaperone networks provides a direct framework for how *daf-21* perturbation sensitizes animals to stress-responsive gene induction during LISA. In parallel, LIN-61, which binds methylated histones and contributes to transcriptional repression[29], likely influences chromatin accessibility and stress-induced transcription, although its precise role during LISA remains to be established.

Beyond cellular and molecular adaptation mechanisms, our work delineates a neural circuit that governs the behavioral reanimation or "awakening" from LISA. AFD sensory neurons and their downstream AIY partners, through PDF neuropeptide and PKA signaling, appear to function as pivotal arousal nodes that promote timely awakening. In contrast, the RIS neuron, known for mediating sleep-like states in *C. elegans*, acts as a suppressive force to delay reactivation. These findings suggest that LISA exit is not solely governed by passive metabolic recovery but involves a regulated neurobehavioral decision underpinned by antagonistic circuit motifs. The cAMP–PKA axis downstream

of AFD-AIY appears to serve as a signaling switch, integrating environmental inputs with internal state to orchestrate behavioral recovery[43,55–57]. This modular and tunable logic mirrors neuroendocrine mechanisms that control sleep–wake transitions in other animals as well as torpor in mammals[67–69], further emphasizing the deep evolutionary conservation of neural strategies that modulate arousal from torpor or dormancy.

Collectively, our findings establish a minimal and genetically tractable system for inducing and dissecting LISA, an extreme state of dormancy, offering a powerful model to investigate the fundamental principles that enable a multicellular life to pause and restart. The identification of core regulators, from HSF-1/DAF-16–dependent transcriptional circuits to auto-lysosomal and neuropeptidergic pathways, provides critical entry points for further mechanistic interrogation and pharmacological modulation. Although SA or hibernation-like states have not been demonstrated in humans, their feasibility remains an open and discussed possibility[11]. Given the growing interest in SA and controlled metabolic suppression for organ preservation, trauma recovery, and deep-space travel[2–11], our work lays the groundwork for future translational exploration of induced stasis in higher organisms.

## Methods

### C. elegans strains
*C. elegans* strains were grown on nematode growth media (NGM) plates seeded with *Escherichia coli* OP50 at 20 °C with laboratory standard procedures unless otherwise specified. The N2 Bristol strain was used as the reference wild type, and the polymorphic Hawaiian strain CB4856 was used for genetic linkage mapping and SNP analysis[70,71]. Forward genetic screens after ethyl methanesulfonate (EMS)-induced random mutagenesis were performed as described previously[72,73]. Approximately 15,000 haploid genomes were screened for *dmaIs8* activators and LISA survivors, yielding at least 12 independent mutants. Mutations were identified by whole-genome sequencing, and complementation was tested by crossing EMS mutants with *daf-21* or *lin-61* heterozygous males. Feeding RNAi was performed as previously described[74]. Transgenic strains were generated by germline transformation as described[75]. Transgenic constructs were co-injected (at 10–50 ng/µl) with dominant *unc-54p::mCherry* or *GFP*, and stable extrachromosomal lines of fluorescent animals were established. Genotypes of mutant or transgenic strains used are as follows: RB1391 *san-1(ok1580) I*, XW9558 *hpo-27(tm5336) I*, VC1669 *aptf-1(gk794) II*, CU5991 *fzo-1(tm1133) II*, EU2959 *vps-15(or1235ts) II; unc-119(ed3) III; orIs21*, KG518 *acy-1(ce2dm) III*, LSC27 *pdf-1(tm1996) III*, LSC39 *pdfr-1(lst34) III*, PY1589 *cmk-1(oy21) IV*, IK800 *gcy-8(oy44) IV*, PR767 *ttx-1(p767) V*, MT150 *egl-3(n150ts) V*, DR47 *daf-11(m47ts) V*, MT14680 *cat-2(n4547) II*, MT18877 *flp-11(n4765) X*, OH15422 *ceh-14(ot900) X*, VC4506 *mtp-18(gk5577[loxP + myo-2p::GFP::unc-54 3' UTR + rps-27p::neoR::unc-54 3' UTR + loxP]) X*, QJ4134 *vps-45(tm246) X; arIs37; jqEx611*, KG744 *pde-4(ce268) II*, AML70 *lite-1(ce314) X; wtfIs5*, GN112 *pgIs2 [gcy-8p::TU#813 + gcy-8p::TU#814 + unc-122p::GFP + gcy-8p::mCherry + gcy-8p::GFP + ttx-3p::GFP]*, JN579 *peIs579 [ttx-3p::casp1 + ttx-3p::Venus + lin-44p::GFP]*, DMS63 *dmaIs8 [hsp-16p::GFP; unc-54p::mCherry] IV*, DMS2974 *dmaEx814 [hsp-16::hsp-16::mStayGold; unc-54p::mCherry]*, DMS124 *dmaIs8 IV; lin-61(dma19)*, DMS125 *dmaIs8 IV; lin-61(dma20)*, DMS407 *dmaIs8 IV; lin-61(dma114)*, DMS2958 *hsf-1(sy441) I; dmaIs8 IV*, KAB122 *louIs8 [ges-1p::nuc-1::mCherry::unc-54 3'UTR]*, RN80 *xmSi01 [mai-2p::mai-2::GFP]*, SJU39 *stjEx24[snb-1p::IlaC22/SL2/dsRed; myo-2p:gfp]*, DCR3055 *wyls629 [gcy-8p::GCaMP6s; gcy-8p::mCherry; unc-122p::gfp]*, DMS3123 *pde-4(ce268) II; pgIs2 [gcy-8p::TU#813 + gcy-8p::TU#814 + unc-122p::GFP + gcy-8p::mCherry + gcy-8p::GFP + ttx-3p::GFP]*, DMS3117 *aptf-1(gk794) II; peIs579 [ttx-3p::casp1 + ttx-3p::Venus + lin-44p::GFP]*, DMS3118 *aptf-1(gk794) II, pgIs2 [gcy-8p::TU#813 + gcy-8p::TU#814 + unc-122p::GFP + gcy-8p::mCherry + gcy-8p::GFP + ttx-3p::GFP]*.

### Plasmid construction and germline transformation
The *Phsp-16.48::hsp-16.48::mStayGold::unc-54 3' UTR* construct was generated by PCR amplification. The *hsp-16.48* promoter and coding sequence were amplified from genomic DNA of wild-type strain N2 using the primer pair 5'-ATCTTCTGGCTTGAACTGCG-3' and 5'-GAAAAGTTCTTCTCCCGTTGAAACGGCTCCATGTTTTGCAACAAAATTAATGGGAATAG-3'. The *mStayGold::unc-54 3' UTR* fragment was amplified from a plasmid (gift from Dr. Mizumoto). The resulting fragments were fused by PCR to generate the full-length *Phsp-16.48::hsp-16.48::mStayGold::unc-54 3' UTR* expression construct.

For transgenic strain generation, the fused PCR product was microinjected at 15 ng µl⁻¹ together with *unc-54p::mCherry* (20 ng µl⁻¹) as a co-injection marker. A 1 kb DNA ladder was added as carrier DNA to a final concentration of 100 ng µl⁻¹. Integrated transgenic lines were generated by UV irradiation (UV Stratalinker 2400, Stratagene) at 600 Lux.

### RNA interference (RNAi)
RNA interference (RNAi) was performed by feeding worms with *E. coli* strain HT115 (DE3) expressing double-stranded RNA (dsRNA) targeting endogenous genes. RNAi clones were obtained from the Ahringer RNAi library. Bacterial clones were streaked onto LB agar plates containing 100 µg ml⁻¹ ampicillin and grown at 37 °C for 16 h. A single colony was inoculated into LB broth containing 100 µg ml⁻¹ ampicillin and cultured overnight at 37 °C. dsRNA expression was induced with 1 mM isopropyl 1-thio-β-D-galactopyranoside (IPTG), and bacteria were seeded onto NGM plates. Plates were incubated at RT for 24 h before use. Developmentally synchronized embryos, obtained by bleaching gravid adults, were placed onto RNAi plates and grown at 20 °C to the indicated developmental stage. Animals were used for imaging, behavioral, or survival assays as indicated.

### Metabolite extraction and GC-MS analysis
*C. elegans* L4 larvae from SA and control conditions were harvested and extracted in cold 90% methanol to achieve a final concentration of 80% methanol per pellet. Samples were incubated at −20 °C for 1 hour, centrifuged at 20,000 × g for 10 minutes at 4 °C, and the resulting supernatants were collected. Quality control (QC) samples were generated by pooling equal volumes of extract from all samples; process blanks contained extraction solvent only. All extracts were dried en vacuo and stored until derivatization. Dried extracts were resuspended in 40 µL of 40 mg/mL O-methoxyamine hydrochloride (MOX) in anhydrous pyridine and incubated at 37 °C for 1 hour. Derivatization was completed by automatic addition of 60 µL N-methyl-N-(trimethylsilyl)trifluoroacetamide (MSTFA + 1% TMCS) and further incubation at 37 °C for 30 minutes. One microliter of the derivatized sample was injected into an Agilent 5977B GC-MS system in split mode (5:1 or 50:1, depending on metabolite abundance). Chromatographic separation was performed on a 30 m Zorbax DB-5MS capillary column (Agilent) with helium as the carrier gas at 1 mL/min. The GC oven temperature was held at 60 °C for 1 minute, ramped at 10 °C/min to 325 °C, and held for 10 minutes.

Mass spectra were acquired using Agilent MassHunter and processed with MassHunter Quant. Metabolites were identified using in-house, NIST, and Fiehn libraries, and peak areas were exported for further analysis. Data were filtered based on %CV ( < 30% in QC samples), QC-to-blank signal ratio (>1.5), and minimal signal intensity (>1000 counts). Statistical analysis was conducted using MetaboAnalystR following log transformation, sum normalization, and Pareto scaling. PCA, heatmaps of the top 25 changing metabolites, and volcano plots were used to compare SA versus the control groups. Metabolites with a fold change >1.5 and raw *P* value < 0.05 were considered significantly altered.

### Metabolite extraction for LC-MS analysis
*C. elegans* L4 larvae subjected to SA or maintained under control conditions were collected, flash frozen, and extracted using a chilled

solution of acetonitrile, water, and methanol (14:4:1 v/v) containing 0.1% ammonium hydroxide, 0.3 μM D9-carnitine, 8 μM D4-succinate, and 2.5 μM uniformly labeled amino acids (Cambridge Isotope Laboratories). Samples were vortexed, homogenized in bead mill tubes, chilled at −20 °C, centrifuged (20,000 × g, 10 min, 4 °C), and supernatants were dried under nitrogen. Extracts were reconstituted in 80% acetonitrile with 0.1% ammonium hydroxide and transferred to autosampler vials for LC-MS/MS analysis.

Metabolites were separated on a Waters BEH zHILIC column using a SCIEX 7600 Zeno-ToF mass spectrometer coupled to an Agilent 1290 Infinity II HPLC system in positive ionization mode. Chromatographic separation employed a gradient of buffer A (25 mM ammonium carbonate in water) and buffer B (99% acetonitrile with 5% water). Data were acquired using high-resolution multiple reaction monitoring (MRM^HR), and chromatograms were integrated using SCIEX Analytics software.

Raw data were drift-corrected using the SERRF algorithm, normalized to the internal standard D9-carnitine, log-transformed, and Pareto-scaled for statistical analysis using MetaboAnalystR. PCA was used to assess overall metabolic differences. Differential metabolite abundance was evaluated using two-sided unpaired Student's $t$ tests with Benjamini–Hochberg false discovery rate correction. Metabolites with adjusted $P < 0.05$ and fold change $> 1.5$ were considered significantly altered. Metabolite identities were confirmed by MS/MS and validated against in-house standards and the Human Metabolome Database (HMDB). Only Level 1 confidence metabolites (confirmed by retention time and reference standard) were included in the final analysis.

### Fluorescence microscopy and imaging

SPE confocal (Leica) and epifluorescence compound microscopes were used to capture fluorescence images. Animals were randomly picked at the same stage and treated with 1 mM levamisole in M9 solution (31742-250MG, Sigma-Aldrich), aligned on a 2% agar pad on a slide for imaging. Identical settings and conditions were used to compare the experimental groups with the control. For quantification of GFP fluorescence, animals were outlined and quantified by measuring gray values using the ImageJ software. The data were plotted and analyzed by using GraphPad Prism.

### RNA sequencing

*C. elegans* N2 animals were maintained at 20 °C. For LISA treatment, synchronized L4 larvae were either subjected to LISA 12 hours or maintained under control conditions prior to collection. Upon sample collection, animals were washed off NGM plates using M9 buffer, collected into 1.5 ml microcentrifuge tubes, and homogenized using a tissue disruptor. Total RNA was extracted using the FastPure Cell/Tissue Total RNA Isolation Kit V2 (RC112-01, Vazyme) according to the manufacturer's instructions. RNA quantity and purity were assessed using a NanoDrop spectrophotometer, and RNA integrity was evaluated by agarose gel electrophoresis. For each sample, 1 μg of total RNA was used for sequencing library construction. Three biological replicates were included for each treatment. Sequencing libraries were prepared and sequenced (paired-end, 150 bp) on the DNBseq platform (Innomics). Differential gene expression analysis was performed using the DESeq2 package. Adjusted $p$ value ≤ 0.05 was used as the threshold to identify the differentially expressed genes.

### *C. elegans* LISA induction, survival and behavioral awakening assays

For the LISA assay, synchronized-stage worms at specific stages were washed off the plate with OP50 as food and collected into a 1.5 mL microcentrifuge tube. For RNAi strains, Day 1 adults (16–24 hours past the L4 stage when RNAi efficiency is sufficient) were used. The worms were washed once with 1 mL of M9 buffer, then the tube was placed on a rack and left to stand at room temperature for the specified duration.

Induction of LISA requires high animal density and was less efficient at LD (low density), defined as fewer than 500 animals per 1 mL isotonic liquid buffer. Unless otherwise specified, animal density for most experiments was ≥ 500 worms/mL. For determining the percentage of LISA, after standing for the designated duration, the M9 supernatant was removed, and worms were transferred to a fresh NGM plate using pipette tips pre-treated with 5% NP-40. After the residual M9 had evaporated, worms were immediately examined under a standard optical microscope for phenotypes, including body length and staging, with L4 characterized by the vulva crescent structure. For assessing survival, after a specific duration of LISA, worms were transferred to a fresh NGM plate and scored for survival after 16 h of recovery. Worms exhibiting a rigid, transparent body and an absence of visible pharyngeal pumping were scored as dead. For assaying awakening, ~100–150 worms after LISA were transferred using a 20 μl tip (prewashed with 5% NP-40) and placed at the center of a 35 mm NGM full-lawn plate. The awakening process was recorded using the WormLab imaging system at room temperature. For acute elevation of cAMP via optogenetic activation, worms were transferred to the center of a 35 mm NGM plate immediately after LISA and exposed to red LED at 2100 lux to induce cAMP production. WormLab recordings were taken every 10 min to monitor the awakening process. All awakening videos were analyzed to determine the percentage of awakened worms, defined as the proportion of animals located outside the original population cluster at 0-60 minutes after transfer.

### Western blot analysis

L4 stage worms were subjected to one of three conditions: normal growth for 12 h, LISA for 12 h with OP50, or recovery for 12 h following 12 h of LISA. The indicated genotypic worms were then washed off the plates and collected using M9 solution. Worm samples (approximately 30 μl animals) were lysed in 1× Laemmli sample buffer (Bio-Rad, #1616737) supplemented with DTT (50 mM; RPI, #D11000-10.0). The samples were mixed thoroughly, boiled at 98 °C for 10 min, and vortexed twice during the incubation. GFP and β-tubulin levels were detected by western blotting using a rabbit anti-GFP monoclonal antibody (MP Biomedicals, #L100036) at a 1:10,000 dilution or a rabbit anti-β-tubulin polyclonal antibody (Proteintech, #10068-1-AP) at a 1:1000 dilution. Blotting membranes (Millipore) were imaged using a ChemiDoc™ Imaging System (Bio-Rad). Band intensities were quantified using ImageJ (NIH). All western blotting experiments were independently repeated at least three times.

### Statistics

Numerical data were analyzed using GraphPad Prism 10 Software (Graphpad, San Diego, CA) and presented as means ± s.e.m. unless otherwise specified, with $P$ values calculated by unpaired two-tailed $t$ tests (comparisons between two groups), one-way ANOVA (comparisons across more than two groups) and two-way ANOVA (interaction between genotype and treatment), with post hoc Tukey and Bonferroni's corrections. The lifespan assay was plotted and quantified using Kaplan–Meier lifespan analysis, and $P$ values were calculated using the log-rank test.

### Reporting summary

Further information on research design is available in the Nature Portfolio Reporting Summary linked to this article.

## Data availability

RNA-seq data have been deposited in the Gene Expression Omnibus under accession number GSE320035. All other data supporting the findings of this study, as well as reagents generated and/or analyzed during the current study, are provided and available from the corresponding author upon reasonable request. Source data are provided with this paper.

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

## Acknowledgements

Some strains were provided by the *Caenorhabditis* Genetics Center (CGC), which is funded by the NIH Office of Research Infrastructure Programs (P40 OD010440), and by Drs. Peter Douglas, Henrik Bringmann, Daniel Colón-Ramos, Noelle L'Etoile, Rosa E. Navarro González, Josh Kaplan, Matt Nelson, and Xiaochen Wang. We also thank the *C. elegans* Reverse Genetics Core Facility (University of British Columbia), National Bioresource Project (Tokyo Women's Medical University), Wormbase.org (NIH grant #U24 HG002223 to P. Sternberg), Wormatlas.org (NIH grant #OD010943 to D.H. Hall.), wormseq.org (Dr. E. O'Rourke), Aging Atlas (Dr. M. Wang) and CenGen for invaluable resources. The work was supported by NIH grants (R35GM139618 to D.K.M.), AHA (24TPA1288391) and 2025 UCSF PBBR New Frontier Research Award (D.K.M.).

## Author contributions

J.L. designed, performed and analyzed most experiments in this work, including characterization of LISA properties, live reporter imaging and LISA phenotypic assays. B.W. made the initial observations on LISA and stress reporter expression and performed EMS screens and genetic mutant mapping. Z.J. and S.W.Y. helped with WGS and imaging analyses. A.B. assisted with data analysis. J.E.C. and J.L.C. performed and analyzed LC-MS and GC-MS experiments. D.K.M. designed and analyzed the *C. elegans* experiments, contributed to project conceptualization, funding acquisition and editing the manuscript. All authors contributed to research materials, project conceptualization and editing manuscript.

## Competing interests

The authors declare no competing interests.
