## [Transparent Peer Review file · Nature Communications]

Induction and regulation of a reversible form of suspended animation in *C. elegans*

Corresponding Author: Dr Dengke Ma

Version 0:

Reviewer comments:

Reviewer #1

(Remarks to the Author)

Animals rely heavily on sensory and stress response pathways to regulate and adapt their metabolism to their environment. One such phenomena is the global, yet reversible, downregulation of metabolic activity termed suspended animation. Like sleep, suspended animation involves an arrest in most movement and consciousness. Unlike sleep, suspended animation is untethered to circadian rhythms while also arresting growth and developmental programming. And unlike sleep, we have a limited understanding of the signaling and molecular mechanisms that regulate and mediate suspended animation. Here, Liu et al use the *C. elegans* system to examine a form of suspended animation that occurs when these nematodes are incubated at high density in isotonic solution (in this case, the standard M9 phosphate saline buffer used for typical *C. elegans* studies). The authors analyze the various parameters required for this form of suspended animation, including time required in solution to induce suspended animation, time to recovery once out of solution, nematode density required for solution, et cetera. They test the stress conditions of starvation, hypoxia, and heat shock to show that these do not mimic this new form of suspended animation. They also test known genetic effectors for response to these stressors, including for dauer diapause arrest. They use RNA-seq and metabolomics approaches to examine gene expression and metabolic changes caused by suspended animation treatment, finding that heat shock chaperones are upregulated, combined with a shift in metabolism towards NAD⁺, ATP, and lactate accumulation. Using a combination of fluorescent reporters and survival assays following suspended animation treatment, the authors screen for mutants with either altered ability to enter suspended animation or altered kinetics of recovery (i.e., awakening) from it. Although they did not identify genes that mediate entry into suspended animation, they did find that DAF-21 and LIN-61 have an antagonistic role towards recovery and survival following suspended animation, as their loss of function mutants maintain Hsp16 gene expression longer and survive better than wild type. Finally, they explore whether key neural circuits and their related signaling molecules regulate exit from suspended animation, and they find that the AFD-AIY neural circuit promotes recovery, whereas the RIS circuit antagonizes recovery.

Overall, the topic of how suspended animation works is quite interesting and of important biological and perhaps even biomedical relevance. The technical approaches employed by the authors are sound, reasonable, and largely well supported with data. However, three major issues limited my enthusiasm for the manuscript in its current form.

First, suspended animation has been studied in *C. elegans* under many different contexts previously, and the background material and the discussion do not give much of the field the justice it deserves. Specifically, it is well known that anoxia (rather than hypoxia) induces suspended animation; indeed, I suspect that the new form of suspended animation that the authors describe is anoxia-induced suspended animation. From experiments using a Clark electrode approach to measure oxygen consumption in solution, it's well established that nematodes in solution will rapidly deplete that solution of oxygen in the time span of minutes to hours. A critical mass of nematodes is required to deplete such oxygen levels in a test tube, which is consistent with the authors' observation that their phenomena is dependent on a high density of animals. Unlike the adaptive changes observed under hypoxia, suspended animation does not require the HIF pathway, similar to the observations from the authors of worm suspended animation post-M9. Like under anoxia, the authors observe mitochondrial fragmentation during their suspended animation paradigm. In short, can the authors demonstrate that this is a novel form of suspended animation? Or have they just found an easier way to study anoxia-induced suspended animation? Either way, they should discuss this issue in the paper and cite the appropriate references.

Second, the authors use a combination of RNA-seq and GFP-based transgenic reporters to show that various HSP-16 genes are regulated by their suspended animation paradigm. Interestingly, the GFP reporters for hsp-16 don't become visible until after the recovery period – not at the end of the M9 incubation period. The authors thus state, "This activation occurs upon re-exposure to permissive environments, indicating that the

stress response is not preemptively deployed during arrest but is part of a reawakening mechanism that buffers the transition from dormancy to active metabolism.” Yet hsp-16 mRNAs are elevated in worms exposed to the standing M9 incubation versus control in the RNA-seq data, which is done immediately at the end of the M9 incubation period, not after the recovery period, showing the transcriptional changes are happening during suspended animation even though those transcriptional changes are not being translated into protein abundance changes until the recovery period. Thus, I don’t agree with the statement that this Hsp16-based gene expression response is part of a reawakening response, as described in the Discussion. Rather, the delay in the appearance of GFP from the transgenic reporters likely either reflects suppressed mRNA translation in the absence of oxygen or perhaps reduced fluorophore maturation in the absence of oxygen (the latter being testable by Western blot)

Third, the authors argue that the recovery from suspended animation is a process that is activity mediated by changes in neural signaling. The authors state, “Beyond cellular and molecular adaptation mechanisms, our work delineates a neural circuit that governs the behavioral reanimation or “awakening” from SA... These findings suggest that SA exit is not solely governed by passive metabolic recovery but involves a regulated neurobehavioral decision underpinned by antagonistic circuit motifs. The cAMP–PKA axis downstream of AFD-AIY appears to serve as a signaling switch, integrating environmental inputs with internal state to orchestrate behavioral recovery.” I’m not sure that I buy that AFD-AIY is acting as a switch in the sense that the circuit has sensed the exit from M9 liquid and now activates a behavioral program to awaken the worm. Rather, couldn’t it just be that the AFD-AIY circuit is robbed of oxygen during M9 incubation and now is simply coming back online with the same level of activity it had before the anoxic exposure? I suppose the authors would counter this point by stating that M9 incubation shuts down the level of AFD::GCaMP fluorescence relative to rab-3p::NLS::GCaMP fluorescence (Figure 4d), suggesting that AFD activity specifically, and no other neurons, is responding to M9 incubation (and subsequent recovery from suspended animation), but the pan-neuronal GCaMP used to measure the activity of the other neurons appears to have an NLS targeting it to the nucleus, which might cause it to experience a different calcium environment compared to the cytosolic AFD::GCaMP. No details were provided regarding the rab-3p::NLS::GCaMP pan-neuronal strain, making it difficult to verify just what this transgenic construct is. Regardless, a single neuron GCaMP control more analogous to AFD::GCaMP would have been more convincing. It would have also been interesting to observe calcium dynamics changes in AIY and RIS, too, given the importance of these neurons highlighted by the authors.

Minor Issues that should be addressed:

It’s unclear what the length of M9 incubation time was for the RNA-seq analysis. This needs to be indicated.

Figure 1c. What does LD stand for? Presumably low density, but there is no discussion of what constitutes low density versus high density, including in the Materials and Methods.

Figure 1g shows that just 3 hours of standing in M9 is enough to induce suspended animation, but the authors don’t try any shorter period to truly map the timing inflexion point.

Fluorescent green text on a white background (e.g., Figure 2e,f) is not easy to read. Given that these are not dual color labels, the authors should change such text to high-contrast black.

Figure 4a lists a short table of genetic strains tested based on their deficiency in various neurons. Unfortunately, strain names rather than genotypes are used to indicate the genetic manipulations, making the graph difficult to decipher, particularly for non-worm researchers not familiar with strain nomenclature. A column should be added that spells out the genotype.

There is not a lot of discussion for the two genes that were isolated based on augmented Hsp16 expression and augmented survival during recovery from suspended animation. For example, there are published roles for DAF-21 and Hsp90 in general for clearing out histones during chromatin remodeling, thereby affecting gene expression. And LIN-61 binds to methylated histones (particularly H4K20me1), where it binds and represses transcriptional activity. Indeed, there is likely a build-up of methylated histones in their suspended animation worms due to the lack of oxygen needed for demethylases to remove those methyl groups. LIN-61 binding to chromatin is probably higher because of that, which likely mediates broad transcriptional repression of genes near such regions of methylated histones, contributing to general suspension of activity as well as the reporter genes for hsp-16. Some discussion of the relevant literature for these genes and how they might be operating would be welcome.

Reviewer #3

(Remarks to the Author)

In this study, the authors demonstrated that culturing nematodes at high density in isotonic M9 solution induces a state of dormancy, characterized by halted development and movement, termed Sustained Animation (SA). SA triggers the expression of heat shock proteins, such as HSP-16, which maintain survival during this state. Additionally, SA alters mitochondrial morphology and affects the accumulation of energy metabolites, which are essential for the resumption of movement post-SA. Changes in lysosomal morphology, which are crucial for survival during SA and the initiation of movement thereafter, were also observed. Furthermore, increased levels of the neuropeptide PDF-1 and its receptor PDFR-1, along with cAMP/PKA in AFD sensory neurons and AIY interneurons, facilitate the initiation of movement following SA. While the phenomenon is potentially intriguing, the lack of precision in the experimental conditions renders the discussion challenging to comprehend. The specific issues are detailed below.

The presence or absence of food in the SA induction experiments should be described more precisely. Food availability is crucial not only for dauer but also for inducing other nematode diapause, such as L1 diapause and ARD (adult reproductive diapause). However, this experiment appears to have been conducted under conditions where OP50 was not sufficiently removed. Without accurately describing this point, it is impossible to discuss the differences between the other types of dormancy. Whether the worms are motile under conditions where OP50 is sufficiently washed off must be precisely determined. Similarly, the meaning of ST+OP50 in Fig. 1j changes entirely depending on whether OP50 was added to a food-free environment or an environment where OP50 was already present.

Although it states that the strain is transferred to an NGM plate during recovery experiments, the presence or absence of OP50 is not clearly explained. The presence or absence of OP50 and any carryover of OP50 must be rigorously discussed in all experiments. Temperature conditions are also critical. While temperature conditions are important for inducing dauer and CID, the SA induction experiment was conducted at room temperature, making it impossible to discuss the effects of temperature. However, because the authors have demonstrated the involvement of temperature adaptation mechanisms such as *hsf-1*, the temperature-responsive neuron AFD, and temperature-responsive genes *gcy-8*, *tx-1*, and *tx-3* in SA induction and recovery, the influence of temperature must be rigorously confirmed and described. Furthermore, the study lacked rigor in terms of worm density during SA induction. While Fig. 1c compares the SA induction efficiency at different densities, subsequent experiments failed to provide the specific worm density used. Worm density should be precisely documented as ≥ 500 worms/mL, not the total worm count used in the experiment.

Line 56:

What exactly are the "agitated (rolling) conditions"?

Fig. 1c:

The study indicates that nematode density is crucial for SA induction, but why is this the case? Is it because physical movement is restricted or because pheromones are secreted? How does this differ from dormancy induced by starvation or low temperature? Did all individuals that did not enter SA develop into young adults? Are phenotypes such as bagging (internal hatching) or individual death not induced? Information on individuals that did not undergo SA is also important.

Fig. 1f:

Starvation during the L1>L2 and L3>L4 stages induces dauer and adult reproductive diapause, respectively. How did you distinguish these from SA?

Fig. 1i:

It has been shown that SA is induced by isotonic M9, but is it also induced by isotonic solutions other than phosphate buffer? If only M9 can induce SA, what is the reason for this?

Fig. 1j:

The legend and methods are insufficiently explained. What does ST in the figure represent? How much OP50, etc., is being added? In addition, why does feeding not promote growth during SA induction?

Ex Fig. 1c:

The authors confirmed the effects of post-SA on lifespan but did not strictly describe the duration of SA. The growth stage that induces SA should be discussed, including information such as temperature and the presence or absence of food.

Fig. 2f in the main text does not correspond to its position in that figure. Figures 2f-i in the figure panel are not referenced in the main text.

Fig. 3g:

The expression of lysosome-related genes *vps-15*, *vps-45*, *rabs-5*, and *hpo-27* should be confirmed during SA induction and recovery to determine whether they are regulated by *hsf-1*, *daf-21*, and *daf-16*.

Fluorescent observation using nematodes: Epistasis between *daf-21* and *hpo-7* should be confirmed by survival during the SA phase in double mutant strains or by double knockdown using RNAi.

Furthermore, *daf-21* regulates dauer formation via the TGF- β pathway together with *daf-11/A* transmembrane guanylyl cyclase, thus requiring discussion of both *daf-11* and the TGF- β pathway.

Heat shock factor-1 intertwines insulin/IGF-1, TGF- β , and cGMP signaling to control development and aging

Janos Barna, Andrea Princz, Monika Kosztelnik, Balazs Hargitai, Krisztina Takacs-Vellai & Tibor Vellai

BMC Developmental Biology volume 12, Article number: 32 (2012)

The explanation for Fig. 3h is not included in the main text.

Many aspects of the connection between these phenomena remain unclear. Does the increase in neuropeptide PDF-1 and its receptor PDFR-1, along with cAMP/PKA, in AFD sensory neurons and AIY interneurons alter HSP gene expression? Fig. 3h indicates that HSP expression affects lysosomal morphology; however, where is the data supporting this? Does the change in metabolite accumulation during SA affect mitochondrial or lysosomal morphology? Is autophagy induction enhanced in the lysosomes during SA?

Fig. 4f, described in the text, is not included in the figures.

Version 1:

Reviewer comments:

Reviewer #1

(Remarks to the Author)

The key technical points I raised in my previous review have largely been addressed.

However, I am still not convinced that this form of suspended animation isn't simply one previously described: anoxia-induced suspended animation (SA). Five hundred worms per mL of M9 rapidly depletes the liquid buffer of oxygen, exposing the worms to anoxia. The fact that regularly dispersing the animals by rolling the microfuge tube blocks this new form of SA induction would suggest that simply aerating the buffer by regularly mixing with the remaining air in the tube is sufficient to restore aerobic levels of oxygen. Indeed, if the model is that there is some sort of quorum-sensing signal released by worms, why would mixing the solution disrupt its function? Yes, the authors provide convincing data that their new SA phenomena is unlikely to be the kind of anoxia-induced SA described by Padilla, Roth, and colleagues, which only occurs in embryos and is SAN-1-dependent. However, there is another form of anoxia-induced suspended animation that occur to larvae and adults (e.g., PMID:24385935) that is dependent on EGL-9, SKN-1, and STL-1, but not HIF-1. This is likely the same phenomenon the authors are observing. I do not see any experiments in the paper that rule out this possibility. Either the authors need to do experiments to show that this new form of SA is not anoxia-induced SA, or they need to seriously discuss this caveat in the Discussion section, including highlighting that they cannot exclude the possibility that these two forms of SA are the same or rely on the same mechanism.

Even if we accept that the form of SA described by the authors is not anoxia-induced SA, all these different forms of SA, each with their own different requirements, raises an important concern: the paper highlights findings as if this new form of SA were generalizable to ALL FORMS OF SA. Indeed, the title (Pausing and restarting life from reversible suspended animation) implies that the mechanism described within is likely broadly generalizable to suspended animation rather than being specific to this new form of suspended animation triggered by a combination of isotonic buffer and high animal density. Indeed, the authors' own experiments demonstrating that this new form of suspended animation is distinct genetically, developmentally, environmentally, and physiologically from most previously described forms of SA clearly show that they are studying something that is fairly narrow and specific rather than generalizable to all forms of SA. To suggest otherwise is to misrepresent the findings and potential impact of the paper; thus, I can't support its publication unless the changes are made to the text that tone down the generalizable nature of their findings:

(1) The title needs to be changed. Perhaps something like: Isosmotic liquid exposure and population density induce a novel reversible form of suspended animation in *C. elegans*.

(2) They need to replace the abbreviation "SA," which suggests applicability to all forms of SA, with something that highlights this specific form of SA. Perhaps "IDSA" for isosmotic and density-induced suspended animation? Alternatively, "LISA" for liquid-induced SA? "SA" can be used in the first two sentences of the abstract where they refer to the general state of SA or other forms of SA induced by other stimuli (e.g., anoxia). It can also be used at lines 33, 40, 42, 53, 64, 69, 75, 80, 266, 273, 276, and 328 in the text for the same reason. But all other mention of SA should indicate that the SA described is specific to this novel form of SA, whether that be with IDSA, LISA, or equivalent abbreviation.

Finally, the authors argue that this new form of SA triggers a drop in neural activity in the AFD neurons (Figure 4d) monitored by an AFD::GCaMP6 transgene, but no such change in other neurons monitored by a Prab-3::NLS::GCaMP6 transgene. I had previously raised the concern that the NLS::GCaMP6 was not the appropriate control for the AFD::GCaMP6 given the former's subcellular localization to the nucleus. The authors responded with a figure meant only for reviewers (Reviewer only Fig. 1) showing no decrease in an RIS::GCaMP reporter, and a decrease in AIY::GCaMP upon SA, similar to that observed in AFD. It's great that they now see a drop in AIY activity. But the AIY::GCaMP pictures in Reviewer only Fig. 1b clearly show that many neurons besides AIY (presumably indicated by arrows, although this was never stated in the description) drop in fluorescence intensity upon SA induction. Thus, it's not just AFD and AIY that show a drop in neural activity under these conditions. Why not just fess up to that point and include this figure in the paper? Indeed, the drop in AIY activity correlates nicely with their demonstrated requirement of AIY in the AIY-ablation experiments (e.g., ttx-3 mutants).

Minor Issues:

I appreciate that the authors have now defined the difference between high density and low density (LD) in the Methods, but why must the reader dig through the methods for something so crucial? The paper would be easier to follow if this was spelled out in the Results (around lines 75-83) and in the legend for Figure 1a-c. For example, exactly how many worms were monitored in the "LD" column of the graph in Figure 1c? What is the control column in this panel? Presumably no liquid immersion at all, but it's not spelled out. Please spell these out, at least in the figure legends.

Lines 245-247. This sentence (To further determine whether SA-induced mitochondrial and lysosomal dynamics we observed might link to awakening from SA, we examined behavioral recovery from SA in mitochondrial and lysosomal mutants) is rather confusing. Did you mean, "To further determine whether THE SA-induced mitochondrial and lysosomal dynamics THAT we observed might BE linkED to awakening from SA?" It needs to be rewritten to make it more clear.

Reviewer #3

(Remarks to the Author)

The authors have adequately addressed the issues I raised.

Version 2:

Reviewer comments:

Reviewer #1

(Remarks to the Author)

My concerns have been nicely addressed. I highly recommend that Reviewer Figure 1 showing that LISA is independent of EGL-9, SKN-1, and HIF-1 being included in the paper (supplement or main figure - either would be fine), as I'm sure I'm not the only one in the field who would have this question. A sentence or two in the Results and Discussion highlighting this point is appropriate, as it further strengthens the finding that LISA is distinct (genetically) from anoxia-induced SA (which should maybe now be called AISA?).

REVIEWER COMMENTS

Reviewer #1 (Remarks to the Author):

Animals rely heavily on sensory and stress response pathways to regulate and adapt their metabolism to their environment. One such phenomena is the global, yet reversible, downregulation of metabolic activity termed suspended animation. Like sleep, suspended animation involves an arrest in most movement and consciousness. Unlike sleep, suspended animation is untethered to circadian rhythms while also arresting growth and developmental programming. And unlike sleep, we have a limited understanding of the signaling and molecular mechanisms that regulate and mediate suspended animation.

Here, Liu et al use the *C. elegans* system to examine a form of suspended animation that occurs when these nematodes are incubated at high density in isotonic solution (in this case, the standard M9 phosphate saline buffer used for typical *C. elegans* studies). The authors analyze the various parameters required for this form of suspended animation, including time required in solution to induce suspended animation, time to recovery once out of solution, nematode density required for solution, et cetera. They test the stress conditions of starvation, hypoxia, and heat shock to show that these do not mimic this new form of suspended animation. They also test known genetic effectors for response to these stressors, including for dauer diapause arrest. They use RNA-seq and metabolomics approaches to examine gene expression and metabolic changes caused by suspended animation treatment, finding that heat shock chaperones are upregulated, combined with a shift in metabolism towards NAD⁺, ATP, and lactate accumulation. Using a combination of fluorescent reporters and survival assays following suspended animation treatment, the authors screen for mutants with either altered ability to enter suspended animation or altered kinetics of recovery (i.e., awakening) from it. Although they did not identify genes that mediate entry into suspended animation, they did find that DAF-21 and LIN-61 have an antagonistic role towards recovery and survival following suspended animation, as their loss of function mutants maintain Hsp16 gene expression longer and survive better than wild type. Finally, they explore whether key neural circuits and their related signaling molecules regulate exit from suspended animation, and they find that the AFD-AIY neural circuit promotes recovery, whereas the RIS circuit antagonizes recovery.

Overall, the topic of how suspended animation works is quite interesting and of important biological and perhaps even biomedical relevance. The technical approaches employed by the authors are sound, reasonable, and largely well supported with data.

Re: Thank you for your overall evaluation and supportive comments.

However, three major issues limited my enthusiasm for the manuscript in its current form.

First, suspended animation has been studied in *C. elegans* under many different contexts previously, and the background material and the discussion do not give much of the field the justice it deserves. Specifically, it is well known that anoxia (rather than hypoxia) induces suspended animation; indeed, I suspect that the new form of suspended animation that the authors describe is anoxia-induced suspended animation. From experiments using a Clark electrode approach to measure oxygen consumption in solution, it's well established that nematodes in solution will rapidly deplete that solution of oxygen in the time span of minutes to hours. A critical mass of nematodes is required to deplete such oxygen levels in a test tube, which is consistent with the authors' observation that their phenomena is dependent on a high density of animals. Unlike the adaptive changes observed under hypoxia, suspended animation does not require the HIF pathway, similar to the observations from the authors of worm suspended animation post-M9. Like under anoxia, the authors observe mitochondrial fragmentation during their suspended animation paradigm. *In short, can the authors demonstrate that this is a novel form of suspended animation? Or have they just found an easier way to study anoxia-induced suspended animation? Either way, they should discuss this issue in the paper and cite the appropriate references.*

Re: Thank you for these insightful and constructive questions. In the original version, we cited reviews and most of the early pioneering work on suspended animation but described the literature only briefly. In the revised version, we have substantially expanded the introduction and discussion on this topic, cited more references, and highlighted the novel aspects of suspended animation in our study in the context of the earlier foundational work by Padilla, Roth and colleagues. We posit that suspended animation in our study is indeed a novel form, which can be induced at multiple larval and adult stages (Fig. 1f) characterized by increased ATP/AMP ratio (Fig. 2i) whereas earlier work using anoxia showed decreased ATP/AMP ratio as expected for *C. elegans* embryo under suspended animation exposed to anoxia (PMID:12006646). Unlike anoxia-induced suspended animation, our scheme is strictly population density dependent yet SAN-1 independent (Fig. 1c, and new Extended Data Fig. 1g). In addition, suspended animation in our study is fundamentally different to the well-established dauer diapause since the latter occurs only during L2-L3 transition, shows distinctly specialized morphologic and physiological changes, and genes required for dauer formation are not required for suspended animation in our study (Extended Data Fig. 1b). Finally, as you also commented, our method of inducing SA is indeed also easier to set up since it does not require environmental anoxia condition, which is strictly required for anoxia-induced embryo suspended animation (PMID:12006646).

Second, the authors use a combination of RNA-seq and GFP-based transgenic reporters to show that various HSP-16 genes are regulated by their suspended animation paradigm. Interestingly, the GFP reporters for *hsp-16* don't become visible until after the recovery period – not at the end of the M9 incubation period. The authors thus state, “This activation occurs upon re-exposure to permissive environments, indicating that the stress response is not preemptively deployed during arrest but is part of a reawakening mechanism that buffers the transition from dormancy to active metabolism.” Yet *hsp-16* mRNAs are elevated in worms exposed to the standing M9 incubation versus control in the RNA-seq data, which is done immediately at the end of the M9 incubation period, not after the recovery period, showing the transcriptional changes are happening during suspended animation even though those transcriptional changes are not being translated into protein abundance changes until the recovery period. Thus, I don't agree with the statement that this Hsp16-based gene expression response is part of a reawakening response, as described in the Discussion. *Rather, the delay in the appearance of GFP from the transgenic reporters likely either reflects suppressed mRNA translation in the absence of oxygen or perhaps reduced fluorophore maturation in the absence of oxygen (the latter being testable by Western blot)*

Re: Thank you for pointing this out. We apologize for the inadequate description in our statement and fully agree that the delay in the appearance of GFP from the transgenic reporters likely reflects suppressed mRNA translation as both the *hsp-16p::GFP* transcriptional and newly generated *hsp-16p::hsp-16::mStayGold* translational reporters showed strongly up-regulated GFP and mStayGold signals only upon exit from suspended animation (new Fig. 2 and Extended Data Fig. 2h). We have revised the text to clarify accordingly. As suggested by this reviewer, we now also provide Western blot results (new Extended Data Fig. 2i-j) showing that HSP-16::mStayGold levels did not increase until recovery, consistent with the imaging data and your interpretation.

Third, the authors argue that the recovery from suspended animation is a process that is activity mediated by changes in neural signaling. The authors state, “Beyond cellular and molecular adaptation mechanisms, our work delineates a neural circuit that governs the behavioral reanimation or “awakening” from SA... These findings suggest that SA exit is not solely governed by passive metabolic recovery but involves a regulated neurobehavioral decision underpinned by antagonistic circuit motifs. The cAMP–PKA axis downstream of AFD-AIY appears to serve as a signaling switch, integrating environmental inputs with internal state to orchestrate behavioral recovery.” I'm not sure that I buy that AFD-AIY is acting as a switch in the sense that the circuit has sensed the exit from M9 liquid and now activates a behavioral program to awaken the worm. Rather, couldn't it just be that the AFD-AIY circuit is robbed of oxygen during M9

incubation and now is simply coming back online with the same level of activity it had before the anoxic exposure? I suppose the authors would counter this point by stating that M9 incubation shuts down the level of AFD::GCaMP fluorescence relative to rab-3p::NLS::GCaMP fluorescence (Figure 4d), suggesting that AFD activity specifically, and no other neurons, is responding to M9 incubation (and subsequent recovery from suspended animation), but the pan-neuronal GCaMP used to measure the activity of the other neurons appears to have an NLS targeting it to the nucleus, which might cause it to experience a different calcium environment compared to the cytosolic AFD::GCaMP. No details were provided regarding the rab-3p::NLS::GCaMP pan-neuronal strain, making it difficult to verify just what this transgenic construct is. Regardless, a single neuron GCaMP control more analogous to AFD::GCaMP would have been more convincing. It would have also been interesting to observe calcium dynamics changes in AIY and RIS, too, given the importance of these neurons highlighted by the authors.

Re: Thank you for these critical and insightful comments. We have included the detailed info about the GCaMP strains used in the Methods, including rab-3p::NLS::GCaMP6s (AML70, lite-1(ce314) X; wtfls5.). Observation of suppressed activity in AFD rather than globally in all neurons upon SA suggests a specific effect on AFD instead of passive and global suppression of oxygen-related metabolism. Though pan neuronal nuclear GCaMP signals reflect cytosolic calcium and help to discern global *C. elegans* neuronal responses, we agree that they are not as ideal as single-neuron GCaMPs to reveal neuronal specificity in responses. Following your suggestions, we monitored the calcium dynamic changes in AIY and RIS in control and SA conditions using previously generated and currently available GCaMP3 strains (made by Bringmann and Bargmann labs). The data indicate that RIS::GCaMP3 did not show apparent changes by SA whereas AIY::GCaMP3 did show similar suppression of activity as AFD::GCaMP during SA and rebound during behavioral recovery (see below Reviewer Only Fig. 1). These results together with those in Fig. 4 support the notion that the neuronal AFD-AIY axis is not only specifically regulated but also essential for behavioral recovery after SA, whereas the RIS neuron might play a permissive role or show more subtle dynamics than AFD-AIY neurons. Given the caveats (response sensitivities and neuronal specificities) of such experiments using early-generation extrachromosomal arrays for RIS::GCaMP3 and AIY::GCaMP3, we remain cautious in interpreting the results and plan to generate and analyze integrated GCaMP6 reporters in follow-up studies. As such, we have also toned down the instructive roles of RIS and AFD-AIY neurons.

Reviewer only Fig. 1 AIY::GCaMP3 but not RIS::GCaMP3 responds to SA. (a) Representative images and quantification of RIS::GCaMP3 responses to SA in L4-stage animals after standing in M9 for 12 hours, showing no significant change. (b) Representative images and quantification of AIY::GCaMP3 responses to SA in L4-stage animals after standing in M9 for 12 hours, showing suppressed and restored AIY::GCaMP3 fluorescence during and after SA, respectively.

Minor Issues that should be addressed:

It's unclear what the length of M9 incubation time was for the RNA-seq analysis. This needs to be indicated.

Re: 12 hours now included.

Figure 1c. What does LD stand for? Presumably low density, but there is no discussion of what constitutes low density versus high density, including in the Materials and Methods.

Re: yes, LD stands for low density, defined as fewer than 500 animals per 1 mL isotonic liquid buffer (Fig. 1c), resulting in incomplete induction of SA, info now included in Method of the revised version.

Figure 1g shows that just 3 hours of standing in M9 is enough to induce suspended animation, but the authors don't try any shorter period to truly map the timing inflexion point.

Re: Shorter than 3 hours makes it more practically difficult to assess developmental arrest in SA based on stage-specific markers and body lengths since unambiguous developmental stage transitions take about 8-10 hours for each larval cycle.

Fluorescent green text on a white background (e.g., Figure 2e,f) is not easy to read. Given that these are not dual color labels, the authors should change such text to high-contrast black.

Re: Corrected.

Figure 4a lists a short table of genetic strains tested based on their deficiency in various neurons. Unfortunately, strain names rather than genotypes are used to indicate the genetic manipulations, making the graph difficult to decipher, particularly for non-worm researchers not familiar with strain nomenclature. A column should be added that spells out the genotype.

Re: Corrected.

There is not a lot of discussion for the two genes that were isolated based on augmented Hsp16 expression and augmented survival during recovery from suspended animation. For example, there are published roles for DAF-21 and Hsp90 in general for clearing out histones during chromatin remodeling, thereby affecting gene expression. And LIN-61 binds to methylated histones (particularly H4K20me1), where it binds and represses transcriptional activity. Indeed, there is likely a build-up of methylated histones in their suspended animation worms due to the lack of oxygen needed for demethylases to remove those methyl groups. LIN-61 binding to chromatin is probably higher because of that, which likely mediates broad transcriptional repression of genes near such regions of methylated histones, contributing to general suspension of activity as well as the reporter genes for hsp-16. Some discussion of the relevant literature for these genes and how they might be operating would be welcome.

Re: Thank you for the suggestion. We have revised the Discussion accordingly.

Reviewer #3 (Remarks to the Author):

In this study, the authors demonstrated that culturing nematodes at high density in isotonic M9 solution induces a state of dormancy, characterized by halted development

and movement, termed Sustained Animation (SA). SA triggers the expression of heat shock proteins, such as HSP-16, which maintain survival during this state. Additionally, SA alters mitochondrial morphology and affects the accumulation of energy metabolites, which are essential for the resumption of movement post-SA. Changes in lysosomal morphology, which are crucial for survival during SA and the initiation of movement thereafter, were also observed. Furthermore, increased levels of the neuropeptide PDF-1 and its receptor PDFR-1, along with cAMP/PKA in AFD sensory neurons and AIY interneurons, facilitate the initiation of movement following SA. While the phenomenon is potentially intriguing, the lack of precision in the experimental conditions renders the discussion challenging to comprehend. The specific issues are detailed below.

The presence or absence of food in the SA induction experiments should be described more precisely. Food availability is crucial not only for dauer but also for inducing other nematode diapause, such as L1 diapause and ARD (adult reproductive diapause). However, this experiment appears to have been conducted under conditions where OP50 was not sufficiently removed. Without accurately describing this point, it is impossible to discuss the differences between the other types of dormancy. Whether the worms are motile under conditions where OP50 is sufficiently washed off must be precisely determined. Similarly, the meaning of ST+OP50 in Fig. 1j changes entirely depending on whether OP50 was added to a food-free environment or an environment where OP50 was already present.

Although it states that the strain is transferred to an NGM plate during recovery experiments, the presence or absence of OP50 is not clearly explained. The presence or absence of OP50 and any carryover of OP50 must be rigorously discussed in all experiments. Temperature conditions are also critical. While temperature conditions are important for inducing dauer and CID, the SA induction experiment was conducted at room temperature, making it impossible to discuss the effects of temperature. However, because the authors have demonstrated the involvement of temperature adaptation mechanisms such as *hsf-1*, the temperature-responsive neuron AFD, and temperature-responsive genes *gcy-8*, *ttx-1*, and *ttx-3* in SA induction and recovery, the influence of temperature must be rigorously confirmed and described. Furthermore, the study lacked rigor in terms of worm density during SA induction. While Fig. 1c compares the SA induction efficiency at different densities, subsequent experiments failed to provide the specific worm density used. Worm density should be precisely documented as ≥ 500 worms/mL, not the total worm count used in the experiment.

Re: Thank you for raising these important considerations. We described the food condition in the Method section and have now revised the text to clarify that OP50 is present in our initial setup for suspended animation (SA) and assays for survival post SA, though SA was robustly induced even after M9 washing, indicating food

independence. We have performed new experiments to show that the absence of OP50, incubation temperature at 15, 20, 25C using precisely temperature-controlled incubators, or incubation in various isotonic buffers does not impact the entry into or exit from suspended animation (new Extended Data Figure 1e-h). We have also revised the text to document the worm density for most experiments as ≥ 500 worms/mL.

Line 56:

What exactly are the “agitated (rolling) conditions”?

Re: The same number of worms in the tube kept on a roller rather than standing still. We changed the word “agitated” to “dispersed” to avoid confusion.

Fig. 1c:

The study indicates that nematode density is crucial for SA induction, but why is this the case? Is it because physical movement is restricted or because pheromones are secreted? How does this differ from dormancy induced by starvation or low temperature? Did all individuals that did not enter SA develop into young adults? Are phenotypes such as bagging (internal hatching) or individual death not induced? Information on individuals that did not undergo SA is also important.

Re: We also find the density requirement intriguing (and fascinating). To address these questions more fully, we have performed experiments showing that the supernatant from suspended animation worms at high density is insufficient to induce suspended animation at low density (new Extended Data Fig. 2i). This would argue against a stable diffusible pheromone that transduced the density signal. We also found that most if not all individuals enter SA at high density while standing in the tube, and individuals that did not enter SA at low density or under rolling/dispersion condition developed into young adults in the presence of OP50 in the tube (Fig. 1j). Unlike starvation, our SA conditions (standing in M9 for 12-16 hrs, in the absence of presence of OP50) did not cause phenotypes such as bagging (internal hatching) or individual death.

Fig. 1f:

Starvation during the L1>L2 and L3>L4 stages induces dauer and adult reproductive diapause, respectively. How did you distinguish these from SA?

Re: Unlike dauer or adult reproductive diapause when cultured on plates with solid media, worms undergoing SA in our paradigm when incubated in isotonic liquid media at high population density do not apparently alter morphologic appearance (Fig. 1b, f). Molecular and metabolic signatures from our transcriptome and metabolomic profiling results (new Fig. 2, Fig. 3 and Supplementary tables), as well as genetic requirements

(*daf-16* and *daf-22* dependency specifically for dauer) are also distinctly different.

Fig. 1i:

It has been shown that SA is induced by isotonic M9, but is it also induced by isotonic solutions other than phosphate buffer? If only M9 can induce SA, what is the reason for this?

Re: Worms standing in S-basal buffer or PBS buffer showed also SA normally (new Extended Data Fig. 1f), indicating robustness of our method inducing SA.

Fig. 1j:

The legend and methods are insufficiently explained. What does ST in the figure represent? How much OP50, etc., is being added? In addition, why does feeding not promote growth during SA induction?

Re: Thank you for pointing this out. Detailed info has been included. ST represents tube “standing” still with animals inside the tube with isotonic buffer. OP50 was washed along with animals into the tube but washed once with M9 before typical SA assays. However, we found the washing step did not affect SA induction, which involves arrested motility and pharyngeal pumping thus feeding did not appear to promote growth. We have performed new experiments to show that the absence of OP50 during and after standing in isotonic liquid, incubation at 15, 20, 25C temperatures using precisely temperature-controlled incubators, or incubation in various isotonic buffers does not impact the entry into or exit from SA (new Extended Data Figure 1e-h).

Ex Fig. 1c:

The authors confirmed the effects of post-SA on lifespan but did not strictly describe the duration of SA. The growth stage that induces SA should be discussed, including information such as temperature and the presence or absence of food.

Re: Info now included: “L4-stage, room temperature and in the presence of OP50.”

Fig. 2f in the main text does not correspond to its position in that figure. Figures 2f-i in the figure panel are not referenced in the main text.

Re: Corrected.

Fig. 3g:

The expression of lysosome-related genes *vps-15*, *vps-45*, *rabs-5*, and *hpo-27* should be confirmed during SA induction and recovery to determine whether they are regulated by *hsf-1*, *daf-21*, and *daf-16*.

Re: We apologize for our inadequate description that led to possible misunderstanding by this reviewer. Mutants for the autolysosome-related genes *vps-15*, *vps-45*, *rabs-5*, and *hpo-27* were used to assess the roles of autophagy and lysosomal fusion processes in promoting survival post SA, not to show that these genes are in the downstream and regulated by HSF-1, DAF-21 or DAF-16. Consistent with published studies, *vps-15*, *vps-45*, *rabs-5*, and *hpo-27* are constitutively expressed and were not identified as direct targets of HSF-1 or DAF-16 (PMID: 27496166; PMID: 25156270). The direct transcriptional HSF-1/DAF-16 target genes responsible for modulating autolysosomal functions and lysosomal tubulation during SA in our paradigm remain to be identified and are among the foci of our follow-up studies.

Fluorescent observation using nematodes: Epistasis between *daf-21* and *hpo-7* should be confirmed by survival during the SA phase in double mutant strains or by double knockdown using RNAi.

Re: We have attempted double RNAi but found efficiency of *daf-21* RNAi with control vector only RNAi is quite limited, without even effectively activating *hsp-16p::GFP*. Thus, we tested whether RNAi against *hpo-27* can suppress enhanced survival in *daf-21* mutants. Percent survival rates of *daf-21*, WT+ *hpo-27* RNAi and *daf-21* + *hpo-27* RNAi showed that *hpo-27* RNAi can partially suppress effects of *daf-21* mutants (Reviewer only Fig. 2), suggesting that DAF-21 recruits HPO-27 dependent and independent mechanisms. Given the caveats of double RNAi and single RNAi in knockdown efficiency and limitation of genetic epistasis in interpreting mechanistic relationship, we have toned down the conclusion on the mechanisms and would like to leave such results out in our manuscript and plan to investigate in depth by follow-up studies.

For reviewer only Fig. 2 Testing genetic epistasis between *daf-21* and *hpo-27*.

Percent survival rates post-SA after standing 36 hrs in M9 showing partial suppression of *daf-21* mutants by *hpo-27* RNAi.

Furthermore, *daf-21* regulates dauer formation via the TGF- β pathway together with *daf-11/A* transmembrane guanylyl cyclase, thus requiring discussion of both *daf-11* and the TGF- β pathway.

Re: Thank you for raising this point. We have tested *daf-11* mutants, which showed normal SA induction (new Extended Data Fig. 1g, h), suggesting the DAF-11/TGF- β pathway is indispensable for dauer formation but not for SA induction. We have included this result and added discussion on this point.

Heat shock factor-1 intertwines insulin/IGF-1, TGF- β , and cGMP signaling to control development and aging

Janos Barna, Andrea Princz, Monika Kosztelnik, Balazs Hargitai, Krisztina Takacs-Vellai & Tibor Vellai

BMC Developmental Biology volume 12, Article number: 32 (2012)

The explanation for Fig. 3h is not included in the main text.

Re: This info is now included.

Many aspects of the connection between these phenomena remain unclear. Does the increase in neuropeptide PDF-1 and its receptor PDFR-1, along with cAMP/PKA, in AFD sensory neurons and AIY interneurons alter HSP gene expression? Fig. 3h indicates that HSP expression affects lysosomal morphology; however, where is the data supporting this? Does the change in metabolite accumulation during SA affect mitochondrial or lysosomal morphology? Is autophagy induction enhanced in the lysosomes during SA?

Re: We apologize for our inadequate description that led to possible misunderstanding by this reviewer. We used the PDF-1, PDFR-1, *gcy-8*, and cAMP/PKA mutants to study processes that regulate awakening post SA, whereas *hsp-16* expression is up-regulated by HSF-1 activation during SA, not post SA. *hsp-16* driven HSP-16 protein increases (new Extended Data Fig. 2) occur post SA as a consequence of restored protein translation. In addition, Fig. 3h shows HSP expression as an indicator of HSF-1 activation, which together with DAF-16 regulate lysosomal morphology (Fig. 3h). We do not claim that HSP expression affects lysosomal morphology given lack of evidence.

Following the reviewer's suggestion, we have also imaged mitochondrial or lysosomal markers upon exogenous metabolites including inositol, succinate and spermidine, the top 3 up-regulated metabolites by GC-MS. Under the conditions tested (1-10 mM supplementation), we did not observe apparent effects as SA had, perhaps due to absorption barriers or limitation of single metabolites at tested doses. We have also analyzed (by imaging and Western blot) the autophagy induction marker ATG-9::GFP

and autophagosome marker LGG-1::GFP but didn't observe striking signal changes, perhaps due to detection threshold/sensitivity of the reporters or alternative autophagy pathway activation. We agree that many aspects of the connection between these phenomena remain unclear and are under our ongoing and future studies. Meanwhile we have revised the manuscript to tone down the unsubstantiated mechanisms and sharpen our scope on the novelty and discovery of these fascinating phenomena.

Fig. 4f, described in the text, is not included in the figures.

Re: This info is now corrected.

REVIEWER COMMENTS

Reviewer #1 (Remarks to the Author):

The key technical points I raised in my previous review have largely been addressed.

However, I am still not convinced that this form of suspended animation isn't simply one previously described: anoxia-induced suspended animation (SA). Five hundred worms per mL of M9 rapidly depletes the liquid buffer of oxygen, exposing the worms to anoxia. The fact that regularly dispersing the animals by rolling the microfuge tube blocks this new form of SA induction would suggest that simply aerating the buffer by regularly mixing with the remaining air in the tube is sufficient to restore aerobic levels of oxygen. Indeed, if the model is that there is some sort of quorum-sensing signal released by worms, why would mixing the solution disrupt its function? Yes, the authors provide convincing data that their new SA phenomena is unlikely to be the kind of anoxia-induced SA described by Padilla, Roth, and colleagues, which only occurs in embryos and is SAN-1-dependent. However, there is another form of anoxia-induced suspended animation that occur to larvae and adults (e.g., PMID:24385935) that is dependent on EGL-9, SKN-1, and STL-1, but not HIF-1. This is likely the same phenomenon the authors are observing. I do not see any experiments in the paper that rule out this possibility. Either the authors need to do experiments to show that this new form of SA is not anoxia-induced SA, or they need to seriously discuss this caveat in the Discussion section, including highlighting that they cannot exclude the possibility that these two forms of SA are the same or rely on the same mechanism.

Response:

We thank the reviewer for raising this important point. We were aware of the suspended animation phenotype described in PMID:24385935 and have cited this study in the Introduction. We agree that anoxia is likely present in our liquid-induced SA paradigm. However, anoxia alone cannot account for several key features of our observations, including SAN-1 independence and stage independence.

To directly address the reviewer's concern, we examined our liquid-induced suspended animation (LISA, see below) by mutants in the EGL-9 pathway previously implicated in anoxia-induced SA. Specifically, we tested *egl-9*, *skn-1*, and *hif-1* mutants (unfortunately, *stl-1* mutants were not available in our laboratory or at the CGC and therefore could not be analyzed). We found that all three mutants entered LISA with penetrance comparable to wild type, with no significant differences in the percentage of animals undergoing developmental and behavioral arrest (Reviewer Figure 1a). Thus, entry into LISA in our paradigm does not require EGL-9, SKN-1, or HIF-1.

Recovery kinetics in *egl-9*, *hif-1*, and *skn-1* mutants were also comparable to wild type, with no significant differences in recovery timing (Reviewer Figure 1b, c). Survival following LISA

was comparable between wild type, *egl-9*, and *hif-1* mutants. In contrast, *skn-1* mutant exhibited mildly improved 24-hour survival (Reviewer Figure 1d). Together, these results indicate that the canonical EGL-9/HIF oxygen-sensing pathway is dispensable for both induction and recovery of LISA under our experimental conditions.

Reviewer Figure 1. **a**, Oxygen-sensing pathways are dispensable for LISA induction and recovery. Percentage of animals entering liquid-induced suspended animation (LISA) under high-density liquid conditions. WT, *egl-9(sa307)*, *hif-1(ia4)*, and *skn-1(zj15)* mutants exhibited comparable LISA penetrance (no significant differences). ($n > 500$ animals per condition, $N > 3$ trials). **b-c**, Time-serial brightfield images and population quantification showing locomotion behavioral recovery post-LISA in wild type and various strains (*egl-9*, *hif-1*, and *skn-1* mutants), indicating recovery timing in *egl-9*, *hif-1*, and *skn-1* mutants was comparable to wild type (no significant differences; p values shown). ($n = 100-150$ worms in each of 3 independent experiments). **d**, Percent survival 24 hours after LISA. Survival rates were comparable between wild type, *egl-9*, and *hif-1* mutants, whereas *skn-1* mutants exhibited significantly improved survival relative to wild type. ($n > 500$ animals per condition, $N > 3$ trials). Data are presented as mean \pm SEM. Statistical analyses were performed using student's two-tailed unpaired *t*-test (**a,d**) or Two-way ANOVA with Bonferroni post-test was used to compare mutant data with WT (**c**).

Together, these data argue that the LISA phenotype described here is mechanistically distinct from the EGL-9–dependent anoxia-induced LISA reported previously. While anoxia may contribute to the physiological context of our assay, LISA does not depend on the canonical EGL-9/HIF pathway. Instead, our findings suggest that LISA engages multiple stress-responsive pathways, including HSF-1 and DAF-16 as described earlier for survival under LISA. We are actively pursuing the regulatory pathway for entry into LISA and mechanistic relationships underlying these observations; however, a detailed dissection of these pathways is beyond the scope of the present study.

Even if we accept that the form of SA described by the authors is not anoxia-induced SA, all these different forms of SA, each with their own different requirements, raises an important concern: the paper highlights findings as if this new form of SA were generalizable to ALL FORMS OF SA. Indeed, the title (Pausing and restarting life from reversible suspended animation) implies that the mechanism described within is likely broadly generalizable to suspended animation rather than being specific to this new form of suspended animation triggered by a combination of isotonic buffer and high animal density. Indeed, the authors' own experiments demonstrating that this new form of suspended animation is distinct genetically, developmentally, environmentally, and physiologically from most previously described forms of SA clearly show that they are studying something that is fairly narrow and specific rather than generalizable to all forms of SA. To suggest otherwise is to misrepresent the findings and potential impact of the paper; thus, I can't support its publication unless the changes are made to the text that tone down the generalizable nature of their findings:

(1) The title needs to be changed. Perhaps something like: Isosmotic liquid exposure and population density induce a novel reversible form of suspended animation in *C. elegans*.

Response:

We thank the reviewer for this thoughtful critique. We agree that our previous title could be interpreted as implying broad generalizability across all forms of suspended animation, which was not our intention. Our goal is to characterize a highly experimentally tractable, completely penetrant, and mechanistically distinct form of reversible SA induced by isosmotic liquid exposure at high population density.

We acknowledge that multiple forms of SA exist in *C. elegans*, each with distinct genetic and environmental requirements. Our own data demonstrate that the LISA described here is genetically, developmentally, and environmentally distinguishable from previously reported paradigms. We agree that it would be inappropriate to imply that the mechanisms identified in this study universally apply to all forms of SA.

To address this concern, we revised the title to: "Induction and regulation of a reversible form of suspended animation in *C. elegans*." This revised title more accurately reflects the scope of our findings without overgeneralization. We have also modified the Abstract and Discussion to clearly state that our conclusions pertain specifically to this liquid- and density-induced SA paradigm. At the same time, we maintain that dissecting the molecular and physiological mechanisms underlying any experimentally tractable form of suspended animation provides important conceptual insight into how animals reversibly pause and resume biological activity. While our findings should not be generalized to all forms of SA, they offer a defined framework for understanding reversible dormancy within this specific context and may inform comparative analyses across different paradigms.

We appreciate the reviewer's guidance in ensuring that the scope and impact of the manuscript are represented accurately and precisely.

(2) They need to replace the abbreviation "SA," which suggests applicability to all forms of SA, with something that highlights this specific form of SA. Perhaps "IDSA" for isosmotic and density-induced suspended animation? Alternatively, "LISA" for liquid-induced SA? "SA" can be used in the first two sentences of the abstract where they refer to the general state of SA or other forms of SA induced by other stimuli (e.g., anoxia). It can also be used at lines 33, 40, 42, 53, 64, 69, 75, 80, 266, 273, 276, and 328 in the text for the same reason. But all other mention of SA should indicate that the SA described is specific to this novel form of SA, whether that be with IDSA, LISA, or equivalent abbreviation.

Response:

We thank the reviewer for this helpful suggestion. We agree that using "SA" throughout may imply broad applicability. To ensure precision, we adopted **LISA (liquid-induced suspended animation)** to refer specifically to the form characterized in this study. The term "SA" is retained only when referring to suspended animation in general or to previously described forms (including in the locations noted by the reviewer). All other instances have been revised to "LISA" throughout the manuscript to clearly define the scope of our findings.

Finally, the authors argue that this new form of SA triggers a drop in neural activity in the AFD neurons (Figure 4d) monitored by an AFD::GCaMP6 transgene, but no such change in other neurons monitored by a Prab-3::NLS::GCaMP6 transgene. I had previously raised the concern that the NLS::GCaMP6 was not the appropriate control for the AFD::GCaMP6 given the former's subcellular localization to the nucleus. The authors responded with a figure meant only for reviewers (Reviewer only Fig. 1) showing no decrease in an RIS::GCaMP reporter, and a decrease in AIY::GCaMP upon SA, similar to that observed in AFD. It's great that they now see a drop in AIY activity. But the AIY::GCaMP pictures in Reviewer only Fig. 1b clearly show that many neurons besides AIY (presumably indicated by arrows, although

this was never stated in the description) drop in fluorescence intensity upon SA induction. Thus, it's not just AFD and AIY that show a drop in neural activity under these conditions. Why not just fess up to that point and include this figure in the paper? Indeed, the drop in AIY activity correlates nicely with their demonstrated requirement of AIY in the AIY-ablation experiments (e.g., *ttx-3* mutants).

Response:

We thank the reviewer for this careful evaluation. The original AIY::GCaMP reporter used in Reviewer-only Fig. 1 was an extrachromosomal array, which exhibited mosaic and variable expression between animals and trials. As a result, signal changes in non-AIY neurons were inconsistently labeled and could be misinterpreted.

To address this concern more rigorously, we have repeated the experiments and generated new datasets with carefully controlled imaging and analysis. The new results confirm that AIY neuronal activity is significantly reduced during LISA induction and recovers upon exit. These revised data are now included in the manuscript (New Extended Data Figure 5c, d).

Importantly, the reduction in AIY activity is consistent with our functional data demonstrating a requirement for AIY (e.g., *ttx-3* mutants) in this process. We agree that these findings strengthen the mechanistic link between neural activity changes and LISA induction, and we have incorporated the revised data accordingly.

Minor Issues:

I appreciate that the authors have now defined the difference between high density and low density (LD) in the Methods, but why must the reader dig through the methods for something so crucial? The paper would be easier to follow if this was spelled out in the Results (around lines 75-83) and in the legend for Figure 1a-c. For example, exactly how many worms were monitored in the "LD" column of the graph in Figure 1c? What is the control column in this panel? Presumably no liquid immersion at all, but it's not spelled out. Please spell these out, at least in the figure legends.

Response:

Corrected. Thanks for the suggestion.

Lines 245-247. This sentence (To further determine whether SA-induced mitochondrial and lysosomal dynamics we observed might link to awakening from SA, we examined behavioral recovery from SA in mitochondrial and lysosomal mutants) is rather confusing. Did you mean, "To further determine whether THE SA-induced mitochondrial and lysosomal dynamics THAT we observed might BE linkED to awakening from SA?" It needs to be rewritten to make it more clear.

Response:

Yes. Corrected.

Reviewer #3 (Remarks to the Author):

The authors have adequately addressed the issues I raised.

Response:

Thank you.

Reviewer #1 (Remarks to the Author):

My concerns have been nicely addressed. I highly recommend that Reviewer Figure 1 showing that LISA is independent of EGL-9, SKN-1, and HIF-1 being included in the paper (supplement or main figure - either would be fine), as I'm sure I'm not the only one in the field who would have this question. A sentence or two in the Results and Discussion highlighting this point is appropriate, as it further strengthens the finding that LISA is distinct (genetically) from anoxia-induced SA (which should maybe now be called AISA?).

Thank you. We agree and have included the EGL-9/HIF-1 results in the Supplementary Figure 1 with appropriate discussion of the mechanisms for LISA distinct to AISA.